# High-throughput *Plasmodium falciparum hrp2* and *hrp3* gene deletion typing by digital PCR to monitor malaria rapid diagnostic test efficacy

**Claudia A Vera-Arias[1], Aurel Holzschuh[1,2], Colins O Oduma[3,4], Kingsley Badu[5], Mutala Abdul-Hakim[5], Joshua Yukich[6], Manuel W Hetzel[2,7], Bakar S Fakih[2,7,8], Abdullah Ali[9], Marcelo U Ferreira[10], Simone Ladeia-Andrade[11], Fabián E Sáenz[12], Yaw Afrane[13], Endalew Zemene[14], Delenasaw Yewhalaw[14], James W Kazura[15], Guiyun Yan[16], Cristian Koepfli[1]***

[1]University of Notre Dame, Notre Dame, United States; [2]Swiss Tropical and Public Health Institute, Allschwil, Switzerland; [3]Kenya Medical Research Institute-Centre for Global Health Research, Kisumu, Kenya; [4]Department of Biochemistry and Molecular Biology, Egerton University, Nakuru, Kenya; [5]Kwame Nkrumah University of Science and Technology, Kumasi, Ghana; [6]Tulane University, New Orleans, United States; [7]University of Basel, Basel, Switzerland; [8]Ifakara Health Institute, Dar es Salaam, United Republic of Tanzania; [9]Zanzibar Malaria Elimination Programme, Zanzibar, Zanzibar, United Republic of Tanzania; [10]University of São Paulo, São Paulo, Brazil; [11]Laboratory of Parasitic Diseases, Fiocruz, Rio de Janeiro, Brazil; [12]Centro de Investigación para la Salud en América Latina, Facultad de Ciencias Exactas y Naturales, Pontificia Universidad Católica del Ecuador, Quito, Ecuador; [13]Department of Medical Microbiology, University of Ghana, Accra, Ghana; [14]Tropical and Infectious Diseases Research Center, Jimma University, Jimma, Ethiopia; [15]Case Western Reserve University, Cleveland, United States; [16]Program in Public Health, University of California, Irvine, Irvine, United States

*For correspondence:
ckoepfli@nd.edu

**Competing interest:** The authors declare that no competing interests exist.

**Abstract** Most rapid diagnostic tests for *Plasmodium falciparum* malaria target the Histidine-Rich Proteins 2 and 3 (HRP2 and HRP3). Deletions of the *hrp2* and *hrp3* genes result in false-negative tests and are a threat for malaria control. A novel assay for molecular surveillance of *hrp2/hrp3* deletions was developed based on droplet digital PCR (ddPCR). The assay quantifies *hrp2*, *hrp3*, and a control gene with very high accuracy. The theoretical limit of detection was 0.33 parasites/µl. The deletion was reliably detected in mixed infections with wild-type and *hrp2*-deleted parasites at a density of >100 parasites/reaction. For a side-by-side comparison with the conventional nested PCR (nPCR) assay, 248 samples were screened in triplicate by ddPCR and nPCR. No deletions were observed by ddPCR, while by nPCR *hrp2* deletion was observed in 8% of samples. The ddPCR assay was applied to screen 830 samples from Kenya, Zanzibar/Tanzania, Ghana, Ethiopia, Brazil, and Ecuador. Pronounced differences in the prevalence of deletions were observed among sites, with more *hrp3* than *hrp2* deletions. In conclusion, the novel ddPCR assay minimizes the risk of false-negative results (i.e., *hrp2* deletion observed when the sample is wild type), increases sensitivity, and greatly reduces the number of reactions that need to be run.

## Editor's evaluation

The study reports the development of high-throughput droplet digital PCR to detect *Plasmodium falciparum* parasites carrying pfhrp2 and pfhrp3 gene deletions. Although there are several PCR-based detection methods already available, the assay is useful as an alternative, particularly in countries and settings where droplet digital PCR is routinely used.

## Introduction

In 2019, over 200 million cases of malaria and over 400,000 deaths were recorded (*World Health Organisation, 2020*). *Plasmodium falciparum* remains the primary cause of malaria in humans. Fast and accurate diagnosis and treatment of clinical episodes are key components of malaria control. Diagnosis is commonly performed either by light microscopy, or rapid diagnostic tests (RDTs). RDTs are lateral flow devices that detect parasite proteins in human blood through immunohistochemistry. Light microscopy requires basic lab infrastructure and skilled microscopists. In contrast, RDTs require minimal infrastructure and training, and results are available within approximately 15 min at a cost of less than 1 USD per test. RDTs are the only field-deployable diagnostic tool available at peripheral health centers and for community screening to diagnose asymptomatic infections, for example through reactive case detection (RCD; *Stuck et al., 2020*). In 2016, over 300 million RDTs were used by malaria control programs (*WHO, 2017*).

The most sensitive RDTs for *P. falciparum* rely on the detection of Histidine-Rich Protein 2 (HRP2) (*Li et al., 2017*). HRP2 is a highly expressed secreted protein, and thus an ideal target for diagnosis. It is also the target for a new generation of ultra-sensitive RDTs with a limit of detection of <100 parasites/μl (*Acquah et al., 2021*; *Hofmann et al., 2019*). While alternative RDTs detecting other proteins, for example parasite lactase dehydrogenase (pLDH), or aldolase, are available, they are generally less sensitive (*World Health Organisation, 2016a*; *World Health Organisation, 2018*). HRP2-based RDTs can also detect HRP3, a structurally similar protein sharing multiple epitopes with HRP2.

In 2010, a report revealed that a large proportion of *P. falciparum* parasites in Peru did not carry the *hrp2* gene (*Gamboa et al., 2010*), and thus could not be detected by HRP2-based RDTs. Since then, an increasing number of reports from Latin America (*Maltha et al., 2012*; *Sáenz et al., 2015*; *Murillo Solano et al., 2015*; *Dorado et al., 2016*; *Akinyi Okoth et al., 2015*), Africa (*Amoah et al., 2016*; *Koita et al., 2012*; *Parr et al., 2017*; *Berhane et al., 2018*; *Menegon et al., 2017*; *Bharti et al., 2016*; *Li et al., 2015*; *Kumar et al., 2013*), and Asia *Johora et al., 2017*; *Pati et al., 2018* found varying proportions of parasites with *hrp2* deletion, reaching up to 80% of clinical cases in certain hospitals in Eritrea (*Berhane et al., 2018*). In addition, *hrp3* can be deleted. The deletion of either *hrp2* or *hrp3*, or both genes, has no known impact on parasite fitness. RDTs can yield a positive result if *hrp2* is deleted but *hrp3* is expressed, but the sensitivity of the RDT is lower in this case (*Kong et al., 2021*).

Molecular surveillance to assess the frequency of *hrp2* and *hrp3* deletion is crucial to decide whether HRP2-based RDTs can be used (*Cheng et al., 2014*). The WHO recommends to use alternative diagnostics if the prevalence of *hrp2* deletion is above 5% (*World Health Organisation, 2016b*). At this level, the number of tests that are false-negative tests because of *hrp2* deletion will exceed the number of tests that are false-negative tests because alternative diagnostics offer a lower sensitivity (*World Health Organisation, 2016b*). The prevalence of *hrp2* deletion has been found to differ substantially within countries (*Murillo Solano et al., 2015*; *Parr et al., 2017*), thus the choice of diagnostics might need to be adapted at subnational level. The feasibility of such approaches depends on the organization of the national control program, barriers to transmission within-country, and other factors. Where low levels of deletions are present, HRP2-based RDTs remain a highly useful tool for diagnosis.

Deletion screening has been classically done using nested PCR (nPCR) followed by gel electrophoresis (*Cheng et al., 2014*). Absence of a band is interpreted as deletion. False-negative results could occur when PCR conditions are suboptimal, or when parasite density is low and amplification is stochastic. To overcome this limitation, threefold repetition of the nested *hrp2* PCR and a control PCR (e.g., *msp2* or *glurp*) is recommended (*Cheng et al., 2014*). As a result, for each sample 12 PCRs need to be run. Deletion status might remain unresolved if results differ among replicates. As an additional problem, multiple clone *P. falciparum* infections are common in most endemic settings (*Lopez and*

*Koepfli, 2021*; *Grignard et al., 2020*). In case of a multiple clone infection with a wild-type parasite and a parasite carrying the deletion, the wild-type parasite will result in a band on the gel when using the nPCR assay. While multiple clone infections will result in a positive RDT (if the density of the wild-type strain is sufficiently high), their presence can mask the presence of deletions, resulting in an underestimation of the frequency of deletion (*Watson et al., 2019*). More recently, quantitative PCR (qPCR) protocols for *hrp2/hrp3* deletion typing were published (*Grignard et al., 2020*; *Schindler et al., 2019*; *Kreidenweiss et al., 2019*), greatly enhancing throughput. However, when parasite densities are low, considerable variation in quantification is observed between replicates (*Koepfli et al., 2016*). While quantification is accurate in the case of clinical samples that are high density, accurate estimates of the population frequency of deletions might require typing of asymptomatic, low-density samples. As a result of the technical challenges for accurate typing, maps of *hrp2* deletion frequency remain scattered and incomplete.

We have developed a novel method for the typing of *hrp2* and *hrp3* deletions based on droplet digital PCR (ddPCR). ddPCR yields highly accurate quantification of parasites (*Koepfli et al., 2016*). In a ddPCR experiment, the reaction volume is partitioned into approximately 15,000 microdroplets, which are then subject to endpoint PCR. Each droplet function as an individual PCR reaction, with amplification occurring if the droplet contains template DNA. ddPCR offers multiple benefits over qPCR for quantification. Template DNA concentration can be derived directly from the number of positive and negative droplets using Poisson statistics. No external standards are required. Quantification is reliable as long as positive and negative droplets are separated, even if amplification is suboptimal, for example in the case of PCR inhibitors. Amplification efficiency, which is crucial for accurate quantification by qPCR, is thus not relevant in a ddPCR experiment. Using two different probes, two targets can be quantified in a single reaction well, for example a control gene and a target gene. Because the target DNA of each assay is partitioned into separate droplets (with the exception of a few droplets partitioned into the same droplet in case of high-density infections), no competition between assays occurs, thus circumventing a common problem of multiplexed qPCR assays. As each droplet with a template can be considered a 'within-well replicate', a single well offers the sensitivity and specificity of a large number of replicates by nPCR or qPCR. The risk of false-negative results (i.e., no amplification when the template is present) is thus minimal compared to nPCR or qPCR. A sample negative for *hrp2/3* and the control gene (e.g., due to pipetting error) will not be classified as carrying a deletion and will be excluded from any calculations on deletion frequency in a population. The novel assay greatly reduces the number of reactions to be run compared to gel-based assays, showed high

**Table 1.** Primer and probe sequences.

| Assay | | Sequence 5'–3' |
|---|---|---|
| | hrp2 exon 2 forward | CATTTTTAAATGCTTTTTTATTTTTATATAG |
| *hrp2* exon 2 | hrp2 exon 2 reverse | CTTGAGTTTCGTGTAATAATCTC |
| | hrp2 exon 2 probe | FAM-CGCATTTAATAATAACTTGTGTAGCAAAAATGC-BHQ-1 |
| | hrp2 exon 1 forward | ATATTTATACATTTTTGTTATTATTTCTTTTTC |
| *hrp2* exon 1 | hrp2 exon 1 reverse | CGTTATCTAACAAAAGTACGGAG |
| | hrp2 exon 1 probe | FAM-CAAAAACGGCAGCGGATAATACTT-BHQ-1 |
| | hrp3 forward | ATGCTAATCACGGATTTCATTTTA |
| *hrp3* | hrp3 reverse | ATCGTCATGGTGAGAATCATC |
| | hrp3 probe | FAM-CCTTCACGATAACAATTCCCATACTTTAC-BHQ-1 |
| | tRNA forward | CATCAAATGAAGATTTAACAAGAG |
| *tRNA* | tRNA reverse | CTTTTTGATTCTATAGTTTCATCTTTATG |
| | tRNA probe | HEX-CTACCTCAGAACAACCATTATGTGCT-BHQ-1 |

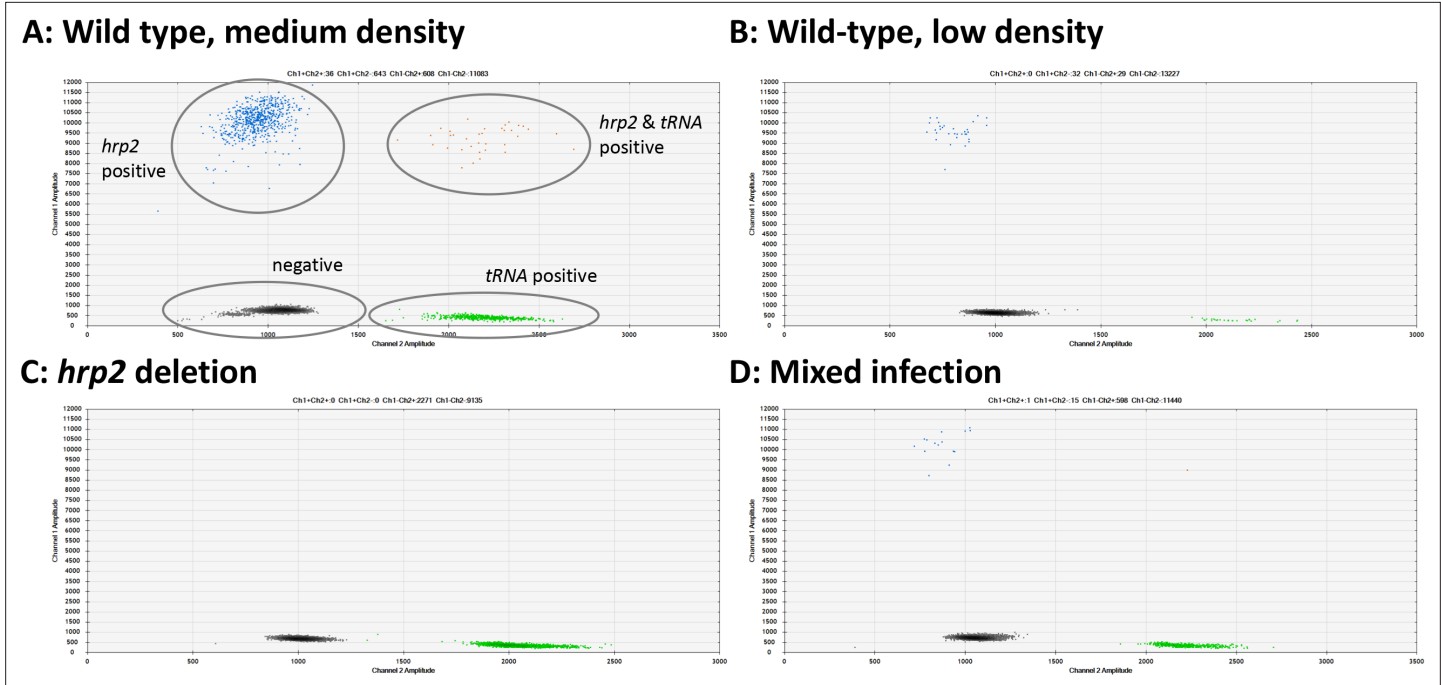

**Figure 1.** Examples of *hrp2* exon 2 deletion typing by droplet digital PCR (ddPCR). Droplets positive for *hrp2* are shown in blue (top left of each panel). Droplets positive for *tRNA* are shown in green (bottom right). Droplets positive for *hrp2* and *tRNA* are shown in orange (top right). Negative droplets (for both *hrp2* and *tRNA*) are shown in gray (bottom left). (**A**) Wild-type infection of medium density (304 parasites/µl, 2 µl DNA used for experiment). Approximately 600 droplets are positive each for *hrp2* and *tRNA*, and 36 for both targets. (**B**) Wild-type sample of low-density (15 parasites/µl): 32 and 29 droplets are positive for *hrp2* and *tRNA*, respectively. (**C**) *hrp2* deletion (1135 parasites/µl): Droplets are positive for *tRNA*, but no droplets are positive for *hrp2*. (**D**) Mixed infection with wild-type parasites and parasites carrying *hrp2* deletion (overall 299 parasites/µl). Only 15 droplets are positive for *hrp2*, but 598 droplets are positive for *tRNA*.

The online version of this article includes the following figure supplement(s) for figure 1:

**Figure supplement 1.** Examples of *hrp3* droplet digital PCR (ddPCR) assays.

**Figure supplement 2.** Examples of *hrp2* exon 1 droplet digital PCR (ddPCR) assays.

sensitivity and accuracy, and can detect the deletion in mixed infections. The assay was extensively validated using culture strains, and field samples from Kenya, Zanzibar/Tanzania, Ethiopia, Ghana, Brazil, and Ecuador.

## Results

### Assay development and validation

New primers and probes were developed for ddPCR assays for *hrp2* exon 1, *hrp2* exon 2, *hrp3*, and *tRNA* (*Table 1*). Upon optimization of the annealing temperature, clear separation between negative and positive droplets was obtained across a wide range of parasite densities. Samples ranging from 1.3 to 49,000 parasites/µl were successfully typed, with samples at densities of >10,000 parasites/µl diluted in $H_2O$. Representative examples are shown in *Figure 1*, *Figure 1—figure supplement 1*, and *Figure 1—figure supplement 2*. No positive droplets for *hrp2*, *hrp3*, or *hrp2* exon 1 were observed in case of deletion, while the separation between negative droplets and those positive for *tRNA* remained clear (*Figure 1C*, *Figure 1—figure supplement 1*, and *Figure 1—figure supplement 2*). In case DNA degradation, loss of DNA during extraction, or no DNA added to the reaction (e.g., pipetting error), no signal will be observed for *tRNA*. In such a case, the sample will be excluded from calculations of *hrp2/3* deletion frequency.

To evaluate the reproducibility and the limit of reliable detection, 248 samples from asymptomatic carriers in Kenya were typed for the *hrp2* exon 2 assay in triplicate. Geometric mean density was 95 parasites/µl, and 47/248 samples were at densities <10 parasites/µl. By ddPCR, 235/248 (94.6%) of samples met inclusion criteria of ≥2 droplets positive for *tRNA* in all three replicates. Highly similar

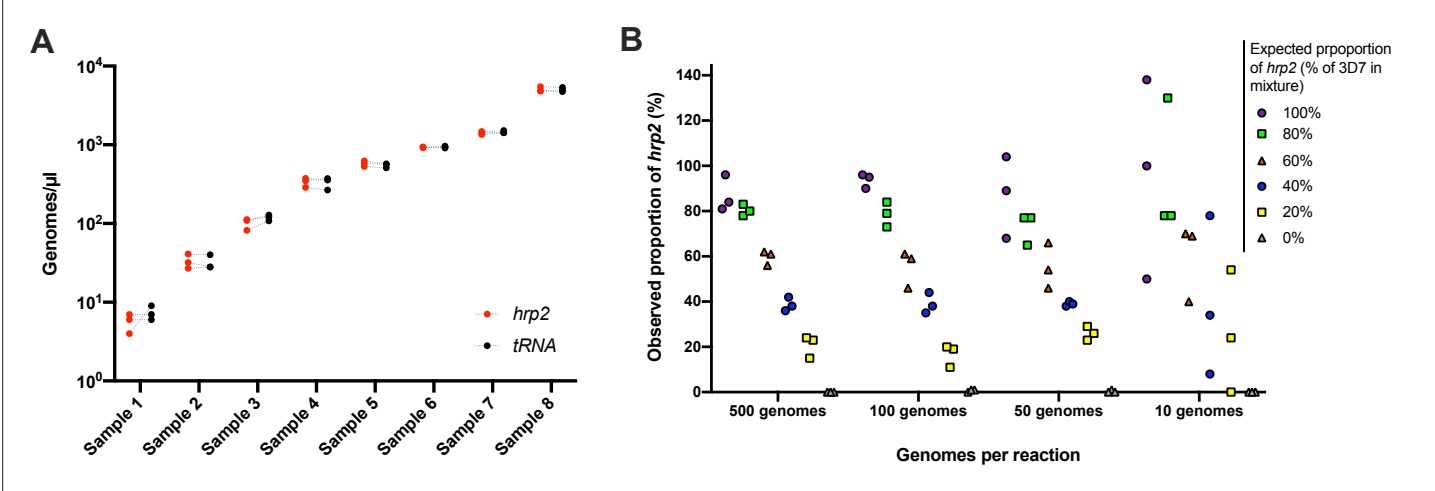

**Figure 2.** Validation of assay. (**A**) Samples typed in triplicate for the *hrp2* exon 2/*tRNA* assay. Representative examples of different parasite densities are shown. For each sample, the quantification of *hrp2* exon 2, and of *tRNA* is shown. Results from the same run are connected by a dashed line. (**B**) Mixtures of 3D7 (wild type) and Dd2 (*hrp2* deletion). Mixtures were run in triplicate at densities of 20–1000 parasites/reaction, and at ratios of 0–100% Dd2. The expected proportion of *hrp2* to *tRNA* copies corresponds to the proportion of 3D7 in the mixture. The observed proportion reflects the expected proportion closely for all mixtures at 1000 and 100 parasites/reaction.

The online version of this article includes the following source data and figure supplement(s) for figure 2:

**Source data 1.** Eight samples typed in triplicate.

**Source data 2.** Mixtures of 3D7 (wild type) and Dd2 (*hrp2* deletion) at different ratios and different concentrations.

**Figure supplement 1.** Mixtures of 3D7 (wild type) and 11,140 (*hrp3* deletion).

quantification among replicates was observed (*Figure 2A*). For each sample, the highest and lowest value of *hrp2* copies/μl was recorded. Correlation among technical replicates was very high ($n = 235$, $R^2 = 0.990$). Likewise, for each sample, the highest and lowest value of *tRNA* copies was recorded, and correlation was very high ($n = 235$, $R^2 = 0.991$). The coefficient of variation (CV) across the triplicates was calculated. CV was 17.7% for the *hrp2* assay, and 17.1% for the *tRNA* assay. As no deletions were observed in this sample set, highly similar quantification of *hrp2* and *tRNA* was expected, and CV between *hrp2* and *tRNA* in the same reaction could be calculated. Mean CV between *hrp2* and *tRNA* was 11.8%. CV was lower among samples of a higher density of >100 parasites/μl ($n = 126$) at 11.5% for hrp2 assay, 10.1% for the tRNA assay, and 4.7% for the within-well *hrp2*/*tRNA* comparison ($n = 126$). The lower CV within wells than among replicates shows that variation in pipetting, resulting slightly different numbers of template added to each replicate reaction, is the main source of variation, while within-well quantification of *hrp2* and *tRNA* is highly accurate.

For the *hrp3*/*tRNA* assay, CV was calculated among 96 triplicates. CV was 19.5% for the *hrp3* assay, and 20.7% for the *tRNA* assay. Mean CV between *hrp3* and *tRNA* across all three replicates was 16.8%.

No deletions were detected in the 248 samples from western Kenya. That is, no samples with droplets for *tRNA* but no droplets for *hrp2* were observed, even though density was low in many samples. In the ddPCR, approximately two-thirds of the reaction volume was transformed into droplets. Applying the threshold of two droplets positive for *tRNA*, three template genomes were required per reaction well to reach that threshold. Up to 9 μl of template DNA could be added to one reaction when primers and probes are kept at 100 μM. Thus, the theoretical limit of detection was 0.33 parasites/μl (three templates in 9 μl of DNA, of which 6 μl are partitioned into droplets). If ≤5 droplets are positive for *tRNA* and a deletion is observed, it is recommended to repeat the sample.

## Detection of mixed infections with *hrp2*- or *hrp3*-negative and wild-type parasites

To test the ability of the assay to detect deletions when only a proportion of all parasites in an isolate carry the deletion, experimental mixtures were prepared with DNA from parasite culture of 3D7 (wild type) and Dd2 (*hrp2* deletion). Mixtures were prepared at ratios from 0% to 100% Dd2, and

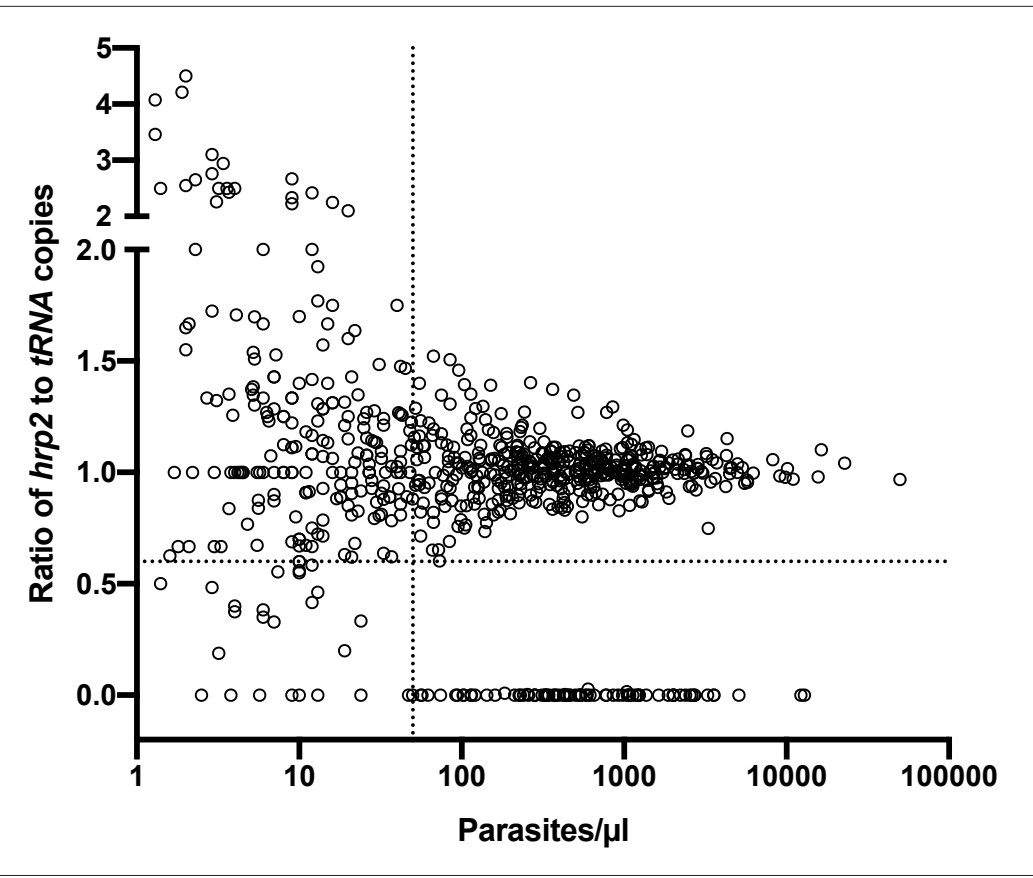

**Figure 3.** Ratios of *hrp2* to *tRNA* copies in 684 field samples. With increasing parasite density (*x*-axis), the ratio becomes close to 1. Deletions (with no wild-type parasites present) have a ratio of 0. Dashed lines show a ratio of *hrp2* to *tRNA* copies of 0.6, and 50 parasites/µl. Mixed infection can be reliably detected at densities >50 parasites/µl, and if >40% of parasites carry the deletion.

The online version of this article includes the following source data and figure supplement(s) for figure 3:

**Source data 1.** Ratios of *hrp2* to tRNA copies in 684 field samples.

**Figure supplement 1.** Ratios of *hrp2* to *tRNA* copies in field samples from Zanzibar (*n* = 91).

**Figure supplement 1—source data 1.** Ratios of *hrp2* to tRNA copies in samples from Zanzibar.

at densities of 10–500 parasites/µl. At densities of 200–1000 parasites/reaction, the quantification by ddPCR represented the mixture ratio with high accuracy (***Figure 2B***). Whenever 40% or more of parasites carried the deletion, the observed ratio was clearly below 100%. In cases where only a small proportion of all parasites carried the deletion (20% Dd2 vs. 80% 3D7), the difference in quantification of *hrp2* and *tRNA* was too small to observe the deletion. Likewise, at densities of 20 parasites/reaction the ratio did not accurately reflect the experimental mixture. Similar results were obtained for mixtures of parasites with *hrp3* deletion and wild type. At densities >20 parasites/reaction, the mixture was reflected with high accuracy (***Figure 2—figure supplement 1***).

The ability to detect a minority clone carrying *hrp2* at a lower frequency (e.g., at 1%) in a sample dominated by a clone carrying the deletion will depend on the overall parasite density. At a density of 1000 parasites/reaction 10 droplets are expected to be positive for *hrp2*, thus the wild-type parasite will be detected. At an overall density of 50 parasites/reaction, <1 droplet is expected to be for *hrp2*, thus the clone will be missed.

The results were corroborated by screening of 739 field samples from five countries. In wild-type samples, a very similar quantification of *hrp2* and *tRNA* was expected. Unless samples carried a clear deletion (i.e., no or very little *hrp2* signal), in all samples with densities of >50 copies/µl, the ratio of *hrp2* to *tRNA* copies was above 0.6 (***Figure 3***). With increasing parasite density, the ratio got closer to 1.

**Table 2.** Comparison between nested PCR and droplet digital PCR (ddPCR)-based *hrp2* deletion typing.

| | Nested PCR | ddPCR |
|---|---|---|
| Total samples | 248 | 248 |
| Met inclusion criteria | 212 (85.5%) | 235 (94.8%) |
| Deletion in 3/3 replicates | 17 | 0 |
| Deletion in 2/3 replicates | 17 | 0 |
| Deletion in 1/3 replicates | 34 | 2 |
| No deletion | 144 | 233 |
| Prevalence of deletion | 8.0% (17/212) | 0% (0/235) |

A lower ratio of *hrp2* to *tRNA* copies was observed in samples from Zanzibar, where the mean ratio of *hrp2*/*tRNA* copies was 0.82 in absence of any samples that carried *hrp2* deletion (*Figure 3—figure supplement 1*). The effect might be caused by sampling and storage procedures. Blood samples from Zanzibar were collected on filter paper, and stored at ambient temperature for over 3 years prior to extraction. Possibly, this could have resulted in DNA degradation, that affected *hrp2* more than *tRNA*.

## Comparison of ddPCR to gel-based nPCR

The *hrp2* exon 2 ddPCR assay was compared to the classical nPCR assay with visualization of products on agarose gel in 248 asymptomatic infections from western Kenya. The density of asymptomatic infections is often low and thus amplification by PCR can be stochastic. All assays were run in triplicate, that is *hrp2* exon 2/*tRNA* by ddPCR, *hrp2* nPCR, and *msp2* nPCR.

Samples were included in the analysis if the PCR for the control gene was positive in all three replicates, that is if >2 droplets were positive for *tRNA* in the ddPCR assay, or a band was detected for *msp2* in all three replicates. For the gel-based assay, 85.5% (212/248) samples met inclusion criteria (*Table 2*). Among those positive, for 144/212 samples a band for *hrp2* was observed in all three replicates. A band in two replicates was observed for 34 samples, and a band in a single replicate for 17 samples. For 17 samples, despite obtaining a band for *msp2* in all three replicates, no band for *hrp2* was observed. These 17 samples would thus be classified as *hrp2* deletion, resulting in a prevalence of deletion of 8.0% (17/212). *Figure 4* shows representative examples of results of the nPCR and ddPCR assays.

By ddPCR, the criteria for inclusion (≥2 droplets for *tRNA*) were met by 94.8% (235/248) of samples. Among them, ≥2 droplets for *hrp2* were detected in all three replicates in 233/235 samples. In two samples in only two of the three replicates ≥2 droplets were positive for *hrp2*. Both of these samples had one replicate with 1 positive droplet for *hrp2*, and five positive droplets for *tRNA*. In none of the samples all three or two out of three replicates were negative for *hrp2*. Thus, the prevalence of deletion by ddPCR was 0%.

## *hrp2* and *hrp3* deletions in Africa and South America

The new ddPCR assay was applied to screen for deletions in 830 samples from Kenya, Zanzibar, Ethiopia, Ghana, Brazil, and Ecuador. The frequency of deletions for all loci and sites is given in *Table 3*.

From Kenya, 241 samples were typed and no deletions of *hrp2* or *hrp3* were observed. In Zanzibar, no deletions were observed among 91 samples. Among 223 samples from Ghana, 1 *hrp2* deletion/wild-type mixed sample was detected (ratio of *hrp2* to *tRNA* copies was <0.6, and parasite density >50 parasites/µl), 1 *hrp3* deletion, and 3 samples with *hrp3* deletion/wild-type mixes.

In Ethiopia, 47 samples met inclusion criteria. One sample carried a deletion of *hrp2* exons 1 and 2, resulting in a frequency of deletion of 2.1%. *hrp3* deletion was observed in 35/47 (74.5%) samples, and one sample carried a mixed infection with wild type/*hrp3* deletion. The sample with *hrp2* deletion was among the samples with *hrp3* deletion, that is both genes were deleted.

From Brazil, 187 samples were screened. Eighty-seven samples carried a deletion of *hrp2*. Two additional samples carried mixed infections with wild-type parasites and *hrp2* deletion. Deletion of *hrp3* was observed in 116/187 (62.0%) samples, and 86/187 (46.0%) samples carried deletions of *hrp2* and *hrp3*. No *hrp2* deletions were observed in Ecuador, but 22/41 (53.7%) samples carried *hrp3* deletions.

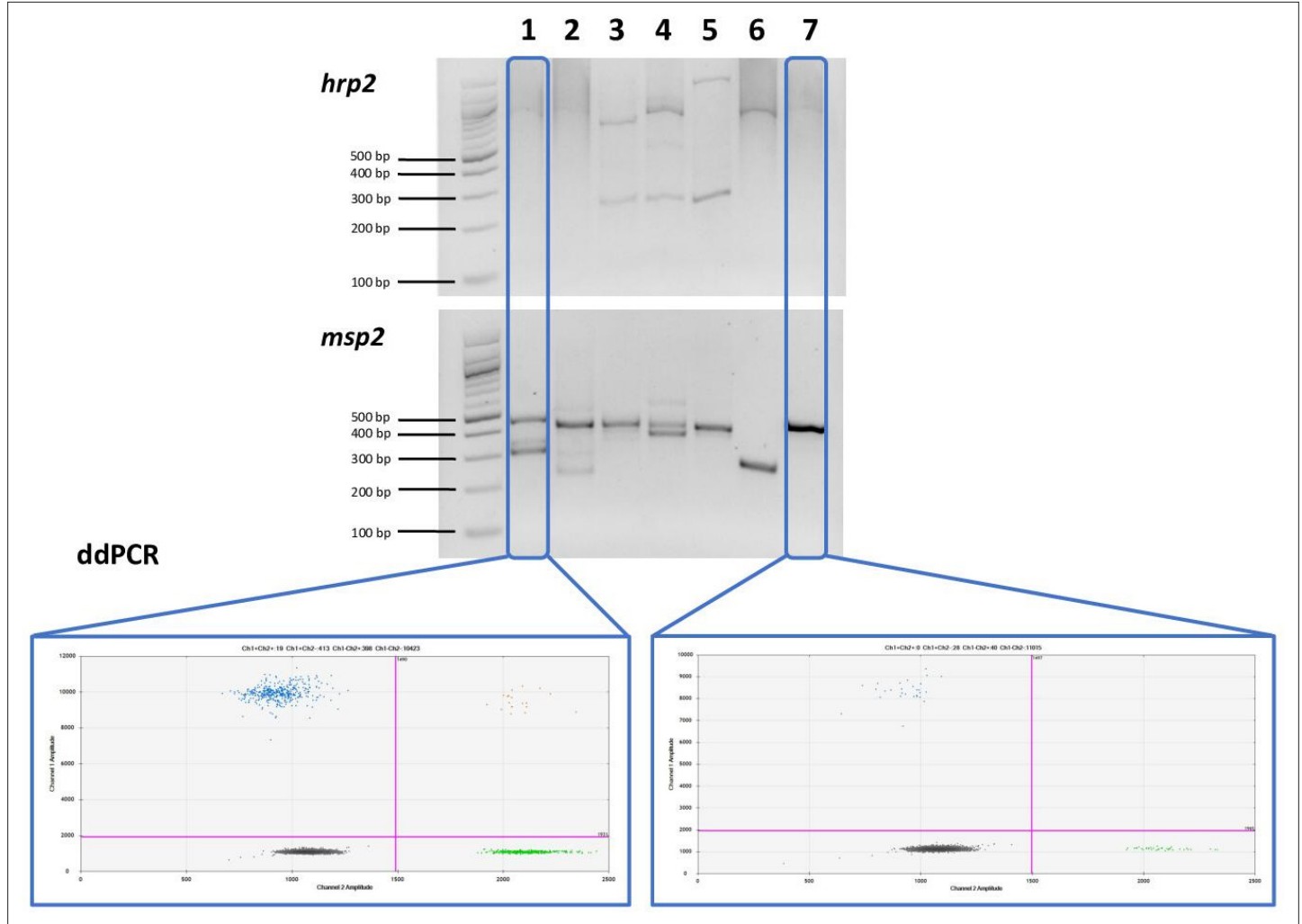

**Figure 4.** Comparison of nested PCR (nPCR) and droplet digital PCR (ddPCR) for *hrp2* deletion typing. Representative examples of results obtained by *hrp2* and *msp2* nPCR, and by ddPCR. The expected size of the *hrp2* band is 228 bp. A band is visible in samples 3, 4, and 5, but no band is visible in samples 1, 2, 6, and 7. *msp2* was run as control for the nPCR assay. *msp2* is a size polymorphic gene with amplicons ranging from approximately 200–500 bp. Bands are observed for all samples, and multiple bands are observed in case of polyclonal infections. L = 100 bp DNA ladder (New England BioLabs). Samples were run in triplicate, and the same results were obtained all three times. By ddPCR, no deletions were observed in any samples. For samples 1 and 7, the *hrp2* exon 2/*tRNA* ddPCR plot is shown. Droplets are visible for both targets, thus no deletion is observed.

## Discussion

Molecular surveillance of the extent of *hrp2* and *hrp3* deletions is a high priority task to select the optimal tools for *P. falciparum* diagnosis. The novel assay based on ddPCR yielded highly accurate results, and was able to detect mixed infections with wild-type parasites and parasites carrying *hrp2* deletion. The good performance of the assay was shown by typing samples from six countries. The samples reflected a range of parasite densities, from low-density asymptomatic infections to high-density clinical infections, and sites represented a range in the frequency of *hrp2* and/or *hrp3* deletions. Across over 800 field samples screened, <10 samples needed to be repeated because of poor separation between negative and positive droplets, or failure of droplet generation.

The assay can be used to screen samples for *P. falciparum* positivity based on the result for *tRNA*, and *hrp2* or *hrp3* deletion in a single assay. Alternatively, it can be used to type samples prior determined *P. falciparum* positive by microscopy or another molecular assay. High sensitivity is required for accurate typing of low-density infections. We validated a threshold of two droplets positive for *tRNA* as the limit of detection. When 9 μl of DNA are used as template, this results in a theoretical limit of

**Table 3.** *hrp2* and *hrp3* deletions in Africa and South America.

| Site | N | *hrp2* exon 2 | *hrp2* exon 1 | *hrp3* | *hrp2* + *hrp3**  | Mixed† |
|------|---|---------------|---------------|--------|------------------|--------|
| | | | | **Deletions** | | |
| Kenya | 241 | 0% (0/241) | 0% (0/241) | 0% (0/241) | 0% (0/241) | |
| Zanzibar | 91 | 0% (0/91) | 0% (0/91) | 0% (0/91) | 0% (0/91) | |
| Ethiopia | 47 | 2.1% (1/47) | 2.1% (1/47) | 74.5% (35/47) | 2.1% (1/47) | 1× *hrp3* |
| Ghana | 223 | 0% (0/226) | 0% (0/223) | 0.4% (1/223) | 0% (0/170) | 1× *hrp2*, 3× *hrp3* |
| Brazil | 187 | 46.5% (87/187) | NA | 62.0% (116/187) | 46.0% (86/187) | 2× *hrp2*, 2× *hrp3* |
| Ecuador | 41 | 0% (0/41) | 0% (0/41) | 53.7% (22/41) | 0% (0/39) | |

*Samples with deletions of *hrp2* and *hrp3*. Note that these samples are also counted as deletions in the columns for *hrp2*, and for *hrp3* (e.g., in Brazil 87 samples carried *hrp2* deletion, of which 86 also carried *hrp3* deletion).

†Samples with only a proportion of all parasites carrying the deletion.

detection of 0.33 parasites/µl. Sensitivity can be further increased by concentrating the DNA prior to typing.

The ddPCR assay showed increased sensitivity and specificity to type low-density infections compared to nPCR. Out of 248 asymptomatic samples from western Kenya, 95% could be analyzed by ddPCR, but only 85% by nPCR. More importantly, results on deletion status differed substantially. By ddPCR, no deletions were observed. By nPCR, no band was observed for *hrp2* in 8% of samples that had a positive band for the control gene (*msp2*) in all three replicates. Thus, the frequency of deletion was above the 5% threshold, and it would be erroneously recommended to discontinue the use of HRP2-based RDTs. The frequency of deletion by nPCR was similar to a previous study conducted in a nearby site in western Kenya that found *hrp2* deletion in 8/89 samples using nPCR (*Beshir et al., 2017*). The direct comparison of ddPCR and nPCR in a large number of samples points to a possible overestimation of *hrp2* deletion frequency by studies relying on nPCR. Further systematic comparison with qPCR-based assays (*Grignard et al., 2020*; *Kreidenweiss et al., 2019*; *Schindler et al., 2019*) will be required to determine the sensitivity and specificity of each assay.

In almost all transmission settings, polyclonal *P. falciparum* infections are frequent (*Lopez and Koepfli, 2021*). Using a gel-based assay for deletion typing, in a mixed infection with a wild-type parasite and a parasite carrying a deletion, the wild-type parasite will produce a band and mask the deletion. These infections can be detected by RDT, but depending on the proportion of polyclonal infections, they can result in pronounced underestimation of the true frequency of deletion (*Watson et al., 2019*). The highly accurate quantification by ddPCR allows detection of mixed infections. Based on experimental mixtures of 3D7 and Dd2, and field samples, mixed infections were reliably detected when at least 40% of parasites carried the deletion, and at densities above 100 parasites per reaction. Using a well-working qPCR assay with an amplification efficacy of 100%, a similar difference between *hrp2* and *tRNA* copy numbers would result in less than half a cycle difference. This is within the normal variation of technical replicates (*Koepfli et al., 2016*; *Hindson et al., 2013*), and thus such mixed infections could not be detected by qPCR.

Twenty-nine samples from Zanzibar and 16 from Ghana had tested negative by HRP2-based RDT despite high density of 100 to >10,000 parasites/µl (*Stuck et al., 2020*), yet no *hrp2* deletions were observed in these populations. False-negative RDT results might be caused due to incorrect handling, prozone effect (*Gillet et al., 2009*), or sequence variation without deletion of the *hrp2* gene (*Nderu et al., 2019*). The finding corroborates the importance of molecular typing. Studies comparing microscopy and RDT results can give important clues for the presence of deletions (*Berhane et al., 2017*), but molecular typing is required for confirmation (*Berhane et al., 2018*).

Though not reported in the literature, deletion of *hrp2* exon 1 only, but not exon 2, could result in a lack of expression of HRP2. In this scenario, the *hrp2* exon 2 assay would not show any deletion, yet HRP2-based RDTs would show a false-negative result. In this study, no discrepancy was observed between deletion status for *hrp2* exons 1 and 2. Thus, for future surveillance, it is recommended to only type samples for *hrp2* exon 2 and *hrp3*. In case of negative RDTs despite no *hrp2* exon 2 deletion, typing for exon 1 can be done.

No *hrp2* deletions were found in Zanzibar, and Kenya, and one mixed infection in Ghana. The data from Ghana contrasts an earlier study that reported a frequency of deletion of >30% (*Amoah et al., 2016*). A single *hrp2* deletion was found in southwestern Ethiopia. This is in stark contrast to very high levels of deletion in western Ethiopia (*Alemayehu et al., 2021*), Eritrea (*Berhane et al., 2018*), and Sudan (*Prosser et al., 2021*). The results corroborate the need for studies assessing *hrp2* deletion and selection of diagnostic tools at subnational level.

Contrasting findings were obtained from the sample sets from South America, with very high levels of deletion in Brazil, and no deletions in Ecuador. Brazil and Ecuador share no borders, and the amount of human migration is limited. The results add to the heterogeneous pattern of *hrp2/hrp3* deletion in South America, with high frequency of *hrp2* deletion in the Amazon (*Gamboa et al., 2010*; *Murillo Solano et al., 2015*; *Góes et al., 2020*), but low levels among the Pacific coast (*Sáenz et al., 2015*; *Murillo Solano et al., 2015*). An outbreak of parasites with *hrp2* deletion at the Peruvian Pacific coast was caused by infections imported from the Amazon (*Baldeviano et al., 2015*). Brazil is committed to *P. falciparum* elimination, and the samples typed originated from the main hotspot of transmission (*Ferreira and Castro, 2016*). RDTs remain relatively little used in Brazil. Microscopy remains the diagnostic method of choice, and RDTs are mostly used in remote areas, for example, populations from Amerindian Reserves and some traditional riverine communities with no access to conventional microscopy. The frequency of *hrp2* deletion in Brazil clearly exceeds the 5% threshold, thus it is recommended that no HRP2-based RDTs are used.

To determine whether the frequency of the deletion exceeds the threshold of 5%, the WHO recommends systematic surveillance and typing of a minimum of 370 per site (*World Health Organization, 2020*). Limitations of the molecular assays available for typing have been a major hindrance to type that number of samples in many sites where *hrp2* or *hrp3* deletion is suspected. Even a low proportion of false-negative results could impact the decision to discontinue HRP2-based RDTs. As a result of the scarcity of field data, the spatiotemporal dynamics of *hrp2* deletion in parasite populations, drivers of the deletion, potential fitness costs, and clinical consequences remain poorly understood. Results from simulation studies suggest that the use of *hrp2*-based RDTs selects for *hrp2*-negative parasites, in particular, if transmission levels are low and a large proportion of all infections turn clinical and result in treatment seeking (*Gatton et al., 2017*; *Watson et al., 2017*). While digital PCR is less common than nPCR or qPCR, it has been successfully established in malaria-endemic countries. The assay is run in 96-well format; thus throughput is comparable to qPCR, and assay setup is similarly straightforward. While costs of digital PCR instruments and reagents are moderately higher than for qPCR, they have declined recently.

In conclusion, the novel, high-throughput, highly sensitive and specific ddPCR assay will facilitate molecular surveillance for *hrp2* and *hrp3* deletion, and thus aid the selection of diagnostic tests to accelerate malaria control and elimination. Data obtained using this assay will help to understand the evolutionary processes underlying the de novo emergence and spread of the deletion.

## Materials and methods

### Digital PCR assays

Four novel ddPCR assays were developed. One assay targets the conserved first 120 bp of *hrp2* exon 2, and thus is located directly adjacent to the histidine-rich repeats. Different breakpoints for the *hrp2* deletion have been described (*Cheng et al., 2014*). The novel primers for exon 2 are located in a region that is deleted in all known deletion variants, thus irrespective of the specific breakpoint, *hrp2* deletion will be detected. The second assay targets *hrp2* exon 1. While exon 1 does not contain antigens that are recognized by RDTs, the deletion of this exon 1 would prevent proper expression of the protein. The third assay targets *hrp3*. The assay amplifies a 101-bp segment in the center of exon 2. Each assay was multiplexed with an assay targeting *serine-tRNA ligase* (PF3D7_0717700, herein referred to as 'tRNA'). *tRNA* is a conserved, essential single copy gene, that is frequently used as reference for gene expression assays (*Friedrich et al., 2014*). In wild-type infections not carrying a deletion, the copy numbers of *tRNA* and *hrp2* or *hrp3* are identical.

Novel primers and probes were developed for all assays (*Table 1*). Assay conditions are given in *Supplementary file 1*. Across over >3000 genomes available through MalariaGen and PlasmoDB, no SNPs were recorded in primer and probe sequences, thus the assay can be used for the screening of

**Table 4.** Field samples included in this study.

| Site | N | Year of collection | Type of diagnosis* | Type of infection | Sample collection method |
|---|---|---|---|---|---|
| Kenya | 248 | 2019 | qPCR | Asymptomatic | Whole blood |
| Zanzibar/Tanzania | 91 | 2017–2018 | qPCR | Asymptomatic | Filter paper |
| Ethiopia | 47 | 2016 | qPCR | Clinical + asymptomatic | Filter paper |
| Ghana | 213 | 2020 | qPCR | Clinical | Whole blood |
| Brazil | 187 | 2010–2013 | Microscopy | Clinical | Whole blood |
| Ecuador | 41 | 2019–2020 | Microscopy | Clinical | Filter paper |

qPCR, quantitative PCR.

*While samples might have been screened by other diagnostic methods, the screening listed was used as inclusion criteria for this study.

samples of global origin. Assay conditions were optimized to achieve maximal separation between positive and negative droplets (*Figure 1*, *Figure 1—figure supplement 1*, *Figure 1—figure supplement 2*).

## 3D7 and Dd2 parasite culture strain mixtures

In order to determine the ability to detect mixed clone infections with only one strain carrying the deletion, experimental mixtures were made from two well-characterized laboratory strains, 3D7 (no deletions), and Dd2 (carrying the *hrp2* deletion). Each strain was cultured separately, DNA extracted, and quantified by ddPCR using the *hrp2/tRNA* ligase assay. No *hrp2* was detected in Dd2. Mixtures were prepared with a concentration (of both strains combined) of 10, 50, 100, and 500 parasites/µl, and with a 3D7 to Dd2 ratio of 100:0, 80:20, 60:40, 40:60, 20:80, and 0:100. Each mixture was run in triplicate using the *hrp2* exon 2/*tRNA* ligase assay. The ratio of *hrp2* to *tRNA* copy numbers was compared to expected values.

## nPCR assay for *hrp2*

In order to compare the ddPCR assay directly to the established nPCR assay, 248 samples from asymptomatic carriers in western Kenya were run by the *hrp2* exon 2 assay and the *hrp2* nPCR in triplicate. Assay conditions for the nPCR followed published protocols (*Akinyi et al., 2013*; *Abdallah et al., 2015*) and are given in *Supplementary file 1*.

## Field samples

Field samples were screened from Kenya, Zanzibar in the United Republic of Tanzania, Ethiopia, Ghana, Brazil, and Ecuador (summarized in *Table 4*). Samples were determined *P. falciparum* positive either by microscopy (Brazil, Ecuador), 18S qPCR (*Rosanas-Urgell et al., 2010* Ethiopia), or varATS qPCR (*Hofmann et al., 2015*; Zanzibar, Kenya, Ghana).

Samples from Kenya (*n* = 248) were from Chulaimbo and Homa Bay in western Kenya close to Lake Victoria. Samples were collected in a cross-sectional survey including individuals of all ages from January to August 2019. Finger-prick samples were collected in EDTA microtainers and infections detected by qPCR. Overall *P. falciparum* prevalence across both sites was 16% (*Oduma et al., 2021*). No diagnosis by RDT or microscopy was done.

Samples from Zanzibar (*n* = 91) had been collected in the frame of a study on RCD from May 2017 to October 2018 (*Stuck et al., 2020*). Asymptomatic infections identified through RCD were typed. During RCD, infections were diagnosed by HRP2/pLDH-based RDT (SD BIOLINE Malaria Ag Pf HRP2/pLHD), a blood spot was collected on filter paper, and infections diagnosed by qPCR (*Stuck et al., 2020*). Prevalence (not including index cases) was 0.8% by RDT and 2.4% by qPCR. All samples with a density (by qPCR) of >100 parasites/µl were selected for *hrp2/hrp3* deletion typing, irrespective of RDT result.

Samples from Ethiopia (*n* = 47) included clinical cases and asymptomatic individuals sampled in June to November 2016 from Jimma Zone, Oromia Region. *P. falciparum* prevalence was 4% by microscopy and 8.3% by qPCR (*Zemene et al., 2018*). No RDT screening was conducted.

From Ghana, two sets of samples were typed. The first set (*n* = 11) was collected in June–September of 2017 from febrile school children aged 5–14 years (*Dieng et al., 2019*). The second set (*n* = 212) was collected in Mankranso and Agona Hospitals in the Ashanti region from febrile patients from September to December 2020.

In Brazil, samples (*n* = 187) were collected in Cruzeiro do Sul, Upper Juruá Valley, northwestern Brazil, in 2010–2013. This is the country's main malaria hotspot, which accounted for nearly 15% of all *P. falciparum* infections in Brazil at the time of the study. Samples were collected from clinical patients 4–73 years of age (mean, 26.6) enrolled for a drug efficacy trial (*Ladeia-Andrade et al., 2016*). Only baseline samples from patients with microscopy- and PCR-confirmed *P. falciparum* infection were included in this study. Even though these samples had been collected nearly a decade ago and might not reflect the current status of *hrp2*/*hrp3* deletion, they were included in light of the known high levels of *hrp2*/*hrp3* deletion in Latin America to confirm the ability of the assays to reliably detect deletions in field samples (*Góes et al., 2020*). Because of little template volume available, Brazilian samples were not screened for *hrp2* exon 1 deletion.

In Ecuador, samples (*n* = 41) were collected from clinical patients from March 2019 to April 2020. The samples were collected in Esmeraldas and Carchi Provinces in the north-west of the country, where most *P. falciparum* cases of Ecuador are reported. All infections were confirmed by microscopy and collected as blood spots in filter paper. Three samples were collected from travelers coming from the Pacific coast in Colombia but diagnosed in Ecuador.

Data are reported as parasites/µl, representing the number of *P. falciparum* genomes per µl eluted DNA. Samples collection and DNA extraction procedures are different among studies (e.g., whole blood vs. filter paper collected, different kits used for extraction). As a result, the conversion from parasites per µl blood to genomes per µl eluted DNA differs among sample sets. For the present study, these differences are not relevant, as for the validation of PCR assays, the number of genomes per µl input DNA is crucial.

## Data analysis

For the analysis of the ddPCR data, the following criteria were applied: A minimum of two droplets positive for *tRNA* were required to include a sample in data analysis. Samples were repeated if a deletion was observed but ≤5 droplets were positive for *tRNA*. If >5 droplets are positive for *tRNA*, the probability of a false-negative result for *hrp2* or *hrp3* (i.e., no positive droplet in a wild-type infection) is less than 1:500. Data for all samples and assays are provided in *Supplementary file 2*. The CV of samples run in replicates, and of different targets within the same tube, was calculated as the standard deviation divided by the mean.

## Acknowledgements

We thank all study participants providing blood samples and the study teams and health center personnel who supported sample collection. We thank Michael T Ferdig and Katelyn M Vendrely for providing culture strain DNA for the 3D7/Dd2 experimental mixtures. Financial Disclosure Statement: This work was supported by NIH grants R21AI137891 awarded to CK, and U19 AI129326, D43 TW001505 awarded to GY (https://www.nih.gov/). The funders had no role in study design, data collection and analysis, decision to publish, or preparation of the manuscript.

## Additional information

### Funding

| Funder | Grant reference number | Author |
| --- | --- | --- |
| National Institutes of Health | R21 AI137891 | Cristian Koepfli |
| National Institutes of Health | D43 TW001505 | Guiyun Yan |

| Funder | Grant reference number | Author |
|--------|------------------------|--------|
| National Institutes of Health | U19 AI129326 | Guiyun Yan |

The funders had no role in study design, data collection, and interpretation, or the decision to submit the work for publication.

## Author contributions

Claudia A Vera-Arias, Data curation, Formal analysis, Investigation, Methodology, Validation, Visualization, Writing - review and editing; Aurel Holzschuh, Colins O Oduma, Kingsley Badu, Mutala Abdul-Hakim, Joshua Yukich, Manuel W Hetzel, Bakar S Fakih, Abdullah Ali, Marcelo U Ferreira, Simone Ladeia-Andrade, Fabián E Sáenz, Yaw Afrane, Endalew Zemene, Delenasaw Yewhalaw, James W Kazura, Guiyun Yan, Investigation, Writing - review and editing; Cristian Koepfli, Conceptualization, Data curation, Formal analysis, Funding acquisition, Investigation, Methodology, Project administration, Resources, Supervision, Validation, Visualization, Writing - original draft, Writing - review and editing

## Author ORCIDs

Aurel Holzschuh  http://orcid.org/0000-0002-2681-1114
Kingsley Badu  http://orcid.org/0000-0002-7886-5528
Joshua Yukich  http://orcid.org/0000-0002-6160-5295
Marcelo U Ferreira  http://orcid.org/0000-0002-5293-9090
Cristian Koepfli  http://orcid.org/0000-0002-9354-0414

## Ethics

Informed written consent was obtained from all study participants or their parents or legal guardians prior to sample collection. The study was approved by the University of Notre Dame Institutional Review Board (approvals 18-08-4803, 19-04-5321, and 18-12-5029), the Institutional Scientific and Ethical Review boards of the Noguchi Memorial Institute of Medical Research, University of Ghana, the Committee on Human Research, Publication and Ethics, School of Medical Science, Kwame Nkrumah University of Science and Technology, Kumasi (CHRPE/AP/375/20), the Zanzibar Medical Research Ethics Committee (ZAMREC/0001/Feb/17), the Institutional Review Board of Tulane University (17-993573), the Institutional Review Board of the Ifakara Health Institute (003-2017), the Ethics Commission of North-western and Central Switzerland (Req-2017-00162), the Institutional Review Board of Institute of Health, Jimma University, Ethiopia (RPGC/486/06), Maseno University Ethics Review Committee (MUERC protocol number 00456), the Ethics Committee for Research in Human Beings of the Pontificia Universidad Católica del Ecuador (CEISH-571-2018), the Ministry of Public Health of Ecuador (MSP-DIS-2019-004-O), and the institutional review board of Oswaldo Cruz Foundation, Brazil (no. 022/2009).

## Decision letter and Author response

Decision letter https://doi.org/10.7554/eLife.72083.sa1
Author response https://doi.org/10.7554/eLife.72083.sa2

# Additional files

## Supplementary files

- Supplementary file 1. Assay conditions.
- Supplementary file 2. Database.
- Transparent reporting form

## Data availability

All data are provided in supplementary file S3.

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
