## [Editor Report]

The study reports the development of high-throughput droplet digital PCR to detect *Plasmodium falciparum* parasites carrying pfhrp2 and pfhrp3 gene deletions. Although there are several PCR-based detection methods already available, the assay is useful as an alternative, particularly in countries and settings where droplet digital PCR is routinely used.

---

## [Decision Letter]

**Decision letter after peer review:**

Thank you for submitting your article "High-throughput *Plasmodium falciparum hrp2* and *hrp3* deletion typing by digital PCR to monitor malaria rapid diagnostic test efficacy" for consideration by *eLife*. Your article has been reviewed by 2 peer reviewers, and the evaluation has been overseen by a Reviewing Editor and Bavesh Kana as the Senior Editor. The reviewers have opted to remain anonymous.

Essential revisions:

Overall, there are concerns regards assay robustness and reliability, further details below:

1. The assay was designed without an internal (human DNA) control. The absence of internal control makes it difficult to decide whether a negative result is true negative or invalid due to error in DNA extraction, pipetting or a poor sample.

2. In the methods section, the authors mentioned the development of four novel ddPCR assays but they have only shown the validation of hrp2 exon2 and tRNA assays in the result section. Figure 1 only shows the performance of hrp2 exon2 and tRNA assays but not hrp2 exon1 and hrp3.

3. The authors used a "wide range" of parasite densities to optimise and assess the reproducibility of the assays. It is not clear what this range is and the data for reproducibility (variation within replicates) such as STD and CV are missing.

4. It is not clear why data for repeatability (assay variation between replicates) is not shown.

5. The accuracy of detection of hrp2 clones in polyclonal infection using ddPCR was not validated by any other method – e.g genome sequencing coverage, qPCR etc. This is important to determine if the test if truly fit for purpose.

6. The authors used absolute quantification to measure DNA for both hrp and tRNA, and then used ratio between the two DNA concentrations to determine hrp2 deletion clone in polyclonal infections. This approach assumes that the amplification efficiency for both assays (hrp and tRNA) is the same or similar, but no data has been presented to show this is the case. This is essential to demonstrate.

Other comments that must be addressed:

1. It is unclear why the authors did not compare the ddPCR against the recently described multiplex qPCR assay approaches. There are three qPCR methods that have been recently published and already deployed in field studies and it is essential that the performance of ddPCR is compared against the qPCR methods, including scalability, ease of use, cost and accessibility, particularly in in resource limited laboratories in endemic countries. The authors should include this in the discussion.

2. Line 80-82: reference is missing.

3. Line 82-84: is it practical to introduce a change in diagnosis policy at sub-national level? Pfhrp2/3-deleted parasites can spread within a country due to test-and-treat policy using HRP2-based RDTs and therefore it is a matter of time before they spread.

4. Line 95: pfhrp2/3 deletions in multiclonal infections were previously reported (Sepulveda et al. 2018) and reference should be included and the authors need to discuss why their methods is better.

5. Line 96-97: Two more qPCR-based assays for the detection of pfhrp2/3 have been published (Lingard et al. 2020 and Kreidenweiss et al., 2019) with comparable sensitivity and good reproducibility, and the authors need to discuss why their assay is better.

6. Line 97-98: It is not clear why the authors suggested quantification is an issue for pfhrp2/3 deletions genotyping. Most HRP2-based RDTs detect as low as 200 parasites per microliter and this parasite density can be accurately detected and quantified using qPCR. The authors need to clarify the issue of pfhrp2/3 deletions genotyping with respect to quantification accuracy.

7. 107-110: This claim was not backed up by data.

8. Line 123: Provide information about which part of exon the hrp3 primers amplify.

9. Line 124: The authors say "Upon optimisation of the assay conditions" – where is the data for assay optimisation?

10. Line 125: The authors mention a wide range of parasite densities but no specific numbers were given.

11. Line 128: Figure 1 – how many parasites per microliter is medium and low density? It is not clear whether the hrp2 assay is based on exon 1 or exon 2. Figure for hrp3 in difference parasite density is also missing.

12. Line 160: How do the authors came up with the 0.33 parasites per microliter?

13. Line 138: Are the 248 samples from Kenya cultured? Otherwise, they should be referred to just as samples rather than isolates.

14. Line 138: The authors used asymptomatic carriers in Kenya to evaluate the reproducibility and limit of detection. Reproducibility and limit of detection are usually conducted using a dilution of a DNA sample with known concentration. The Kenyan samples should be used for validation of the ddPCR assays. In fact, the authors correctly referred the data in figure 2 as "validation". How were the genome copies measured for the Kenyan samples originally and what method was used to quantify them?

15. Line 139-140: most pfhrp2/3 genotyping assays (nPCR and qPCR) are evaluated based on parasite density (parasites per microliter) and it is difficult to compare ddPCR performance if the parameter is genomes per microliter. The authors need to either use parasite density throughout the manuscript or include a genome to parasite density conversion formula in the analysis section of the methods.

16. Line 139: again the validation was for hrp2 exon-2 and the validation for hrp2 exon-1 and hrp3 are missing.

17. Line 164: For validating deletions in mixed clones the authors used parasite culture lines 3D7 and Dd2. Why have not the authors used the same parasite lines to assess the reproducibility, repeatability and limit of detection of all individual assays (hrp2 exon-1, exon2, hrp3 and tRNA)?

18. Line 164: Clone mixture was done for hrp2 but not for tRNA and hrp3. Since identification of hrp2/3-deleted clone in a multiclonal infection is relative to tRNA detection of mixture should also be validated using tRNA assay.

19. Line 270: not clear how 0.33 parasites per microliter is determined.

20. Line 293: This depends whether the quantification is relative or absolute. In a relative quantification, both targets with similar amplification efficiency are expected to have similar variation of technical replicates and therefore not affecting the mean cycle/quantification threshold value. The authors need to be address this.

21. Line 296: also due to low parasite density.

22. Line 306: There will be operational challenges to suggest selection of diagnostic tools at sub-national level. Once deletion arise locally it is a matter of time before they spread nationally due movement of people and test-and-treat policy using RDT.

23. Line 318: systematics surveillance of hrp2/3 deletions should be recommended (if it has not been done yet) before diagnostic policy change.

24. Line 331: Authors mention hrp2 and hrp3 in the conclusion but presented no data about the performance of ddPCR for hrp3.

25. Line 356: The authors stated "The novel primers for exon-2 are located in a region that is deleted in all known deletion variants, thus irrespective of the specific breakpoint, hrp2deletion will be detected" but then they included an assay that targets hrp2 exon 1. Isn't targeting exon-2 only enough if exon-2 is always deleted? This need to be clarified in the materials section as well as discussion.

26. Line 359-360: what part of gene does the hrp3 primers target?

27. Line 366-367: Hrp2 probe mutations – Ghana and Mali samples (in the genome database) carry mutation at 5’ end of the probe (C to T). Hrp2 reverse primer targets a region with the same sequence as hrp3 with one nucleotide sequence difference and it is surprising that the primer doesn’t amplify hrp3. In qPCR assay published by Grignard et al. 2020, hrp2 reverse primer with three nucleotide differences amplifies hrp3 and the authors introduced a deliberate mutation at the 3’ end to increase specificity. Vera-Arias and colleagues need to explain how they did not get non-specific amplification (signal) in the hrp3 channel.

28. Line 377: why was hrp3 clone detection in mixture strains not done?

29. Line 380: We know that clone mixtures occur at larger ratio difference such as 1:100 and 1:1000 – have the authors considered this? The parasite density used for the mixture is very low (if 500 genome is equal to 500 parasite density per microliter).

30. Line 381: how was the ratio calculated? Is it based on standard or relative quantification?

32. Line 384: why was the ddPCR not compared to published qPCR assays?

33. Line 429: analysis method for quantification of parasite density (genome) per microliter is missing.

34. Line 36 – a "threat" for malaria control.

35. Line 39 – please clarify if LOD parasite/uL is referring to per ul of extract, PCR reaction or original input sample (e.g. whole blood)?

36. Line 60 – how effective are RDTs for diagnosing asymptomatic infections for malaria? They are probably not that sensitive, particularly in the case of COVID-19 detection for asymptomatic cases. Is this the same for malaria?

37. Line 80-82 "At this level, the number of false-negative…." – suggest re-write of this sentence as it is confusing to readers what the authors are intended to compare.

38. Line 87-88 "False-negative results could occur when PCR conditions are suboptimal, or when parasite density is low and amplification is stochastic." – a good nPCR assay can potentially amplify up to Ct35. If one can't detect the deletion via nPCR, is the deletion number significant enough? It is unlikely to be picked up by RDTs either at this load. Perhaps this can be confirmed by real-time or digital PCR but how significant is this low parasite load? If the deletion of these genes do not have impact on parasite fitness, then will this low level be of any concern?

39. Line 92-04 – will multiple clone (mixed) infection also showed up in RDTs? Might be a good point to add RDTs false negative here to highlight the importance of a good molecular method to detect mixed infections.

40. Line 166 – "part of all parasites"…the authors meant "a proportion of parasites"?

41. Line 171-174 – in cases where the deletion is hard to observe or densities are too low (small amount of deletion comparing to wild type), do the authors suggest considering running ddPCR in single assay format rather than multiplex to eliminate any resource competitions that might occur in a multiplex reaction?

42. Line 268 – "An assay" with high sensitivity is required for accurate typing…

43. Line 275-276 – no deletions were observed by ddPCR but 8% were negative hrp2 but positive for msp2 in nPCR testing. What was done to verify that it was not due to false negative by ddPCR?

44. In Discussion section, it might be useful to mention the cost per sample for ddPCR comparing to nPCR as authors mentioned the reduction in the number of reactions to be run earlier in the article. Cost-effectiveness can usually be a key determinant for many labs to consider when running studies like this.

45. Table 1 and supplementary file S1: add quencher (e.g. BHQ1) for Taqman probe sequences.

46. Line 336 -435 – Methods section is difficult to follow. Suggest re-arrange of the sections into Isolates (highlighting how these isolates were determined as positive infections [e.g. by qPCR, microscopy or RDT?]), parasite culture mixtures, ddPCR, nPCR and then data analysis. Since there are many field isolates from different origins, perhaps a table summarising these isolates and positivity rates, how they were detected and special notes on the isolates and how the authors use each isolate to characterise their droplet digital PCR assay. This will make it easier for readers to follow.

47. In Supplementary file, please clearly specify the droplet digital PCR instruments, mastermix used, primers and probe (final or working concentrations).

48. What are the pre-PCR processing for all various field isolates? Were the DNA extracted using different kits and how were they extracted? Please add in manuscript or supplementary methods.

49. The author also mentioned that this is a validated ddPCR assay. Please consider adding a PCR amplification (droplet scatter) plot showing positive/negative partition with threshold and information on some QC data (for example, positive/negative controls used, acceptable droplet numbers, any rain drop issues from mixed infections etc).

*Reviewer #1 (Recommendations for the authors):*

The study from Vera-Arias and colleagues makes an important contribution in that it reports the development of high-throughput droplet digital PCR (ddPCR) to detect *Plasmodium falciparum* with pfhrp2 and pfhrp3 gene deletions. Though there are several nPCR and qPCR-based detection methods already available, the assay is useful as an alternative, particularly in countries and settings where ddPCR is routinely used. This has the potential to assist in surveillance for pfhrp2/3 deletions programs where RDT designed to detect HRP2 are the primary test leading to false negative results, particularly in medium to high transmission settings.

The data presented do not give this reviewer confidence that the quantitative aspects of the ddPCR assay used here are robust and reliable.

1. Assay was designed without an internal (human DNA) control. The absence of internal control makes it difficult to decide whether a negative result is true negative or invalid due to error in DNA extraction and pipetting.

2. In the methods section, the authors mentioned the development of four novel ddPCR but they have only shown the validation of hrp2 exon2 and tRNA assays in the result section. Figure 1 only shows the performance of hrp2 exon2 and tRNA assays but not hrp2 exon1 and hrp3.

3. The authors used a "wide range" of parasite densities to optimise and assess the reproducibility of the assays. It is not clear what this range is and the data for reproducibility (variation within replicates) such as STD and CV are missing.

4. It is not clear why data for repeatability (assay variation between replicates) is not shown.

5. The accuracy of detection of hrp2 clones in plyclonal infection using ddPCR was not validated by any other method – e.g genome sequencing coverage, qPCR etc.

6. The authors used absolute quantification to measure DNA for both hrp and tRNA, and then used ratio between the two DNA concentrations to determine hrp2 deletion clone in polyclonal infections. This approach assumes that the amplification efficiency for both assays (hrp and tRNA) is the same or similar, but no data has been presented to show this is the case.

Other comments and Corrections

It is unclear why the authors did not compare the ddPCR against the recently described multiplex qPCR assay approaches. There are three qPCR methods that have been recently published and already deployed in field studies and it is essential that the performance of ddPCR is compared against the qPCR methods, including scalability, ease of use, cost and accessibility, particularly in in resource limited laboratories in endemic countries. The authors should include this in the discussion.

*Reviewer #2 (Recommendations for the authors):*

Overall, the content of this paper is informative and can be of interest to *eLife* readers but the manuscript needs to be polished. Some sections of the paper (for example – methods) may require re-structure and overall writing can be more concise as some sentences are a bit awkward and confusing to read.

Line 36 – a "threat" for malaria control.

Line 39 – please clarify if LOD parasite/uL is referring to per ul of extract, PCR reaction or original input sample (e.g. whole blood)?

Line 60 – how effective are RDTs for diagnosing asymptomatic infections for malaria? My understanding is that they are not that sensitive, particularly in the case of COVID-19 detection for asymptomatic cases. Is this the same for malaria?

Line 80-82 "At this level, the number of false-negative…." – suggest re-write of this sentence as it is confusing to readers what the authors are intended to compare.

Line 87-88 "False-negative results could occur when PCR conditions are suboptimal, or when parasite density is low and amplification is stochastic." – a good nPCR assay can potentially amplify up to Ct35. If you can't detect the deletion via nPCR and then is the deletion number significant enough? It is unlikely to be picked up by RDTs either at this load. Perhaps this can be confirmed by real-time or digital PCR but how significant is this low parasite load? If the deletion of these genes do not have impact on parasite fitness, then will this low level be of any concern?

Line 92-04 – will multiple clone (mixed) infection also showed up in RDTs? Might be a good point to add RDTs false negative here to highlight the importance of a good molecular method to detect mixed infections.

Line 166 – "part of all parasites"…you meant "a proportion of parasites"?

Line 171-174 – in cases where the deletion is hard to observe or densities are too low (small amount of deletion comparing to wild type), will you be considering running ddPCR in single assay format rather than multiplex to eliminate any resource competitions that might occur in a multiplex reaction?

Line 268 – "An assay" with high sensitivity is required for accurate typing…

Line 275-276 – no deletions were observed by ddPCR but 8% were negative hrp2 but positive for msp2 in nPCR testing. What did you do to verify that it was not due to false negative by ddPCR?

In Discussion section, it might be useful to mention the cost per sample for ddPCR comparing to nPCR as authors mentioned the reduction in the number of reactions to be run earlier in the article. Cost-effectiveness can usually be a key determinant for many labs to consider when running studies like this.

Table 1 and supplementary file S1: don't forget to add quencher (e.g. BHQ1) for Taqman probe sequences.

Line 336 -435 – Methods section is difficult to follow. Suggest re-arrange of the sections into Isolates (highlighting how these isolates were determined as positive infections [e.g. by qPCR, microscopy or RDT?]), parasite culture mixtures, ddPCR, nPCR and then data analysis. Since there are many field isolates from different origins, perhaps a table summarising these isolates and positivity rates, how they were detected and special notes on the isolates and how the authors use each isolate to characterise their droplet digital PCR assay. This will make it easier for readers to follow.

In Supplementary file, please clearly specify the droplet digital PCR instruments, mastermix used, primers and probe (final or working concentrations).

What are the pre-PCR processing for all various field isolates? Were the DNA extracted using different kits and how were they extracted? Please add in manuscript or supplementary methods.

The author also mentioned that this is a validated ddPCR assay. Please consider adding a PCR amplification (droplet scatter) plot showing positive/negative partition with threshold and information on some QC data (for example, positive/negative controls used, acceptable droplet numbers, any rain drop issues from mixed infections etc).

[Editors’ note: further revisions were suggested prior to acceptance, as described below.]

Thank you for resubmitting your work entitled "High-throughput *Plasmodium falciparum hrp2* and *hrp3* deletion typing by digital PCR to monitor malaria rapid diagnostic test efficacy" for further consideration by *eLife*. Your revised article has been evaluated by Bavesh Kana (Senior Editor) and a Reviewing Editor.

Whilst reviewers concurred that the work was improved, there were still some remaining concerns that need to be addressed. These are outlined below.

Essential revisions:

1. In response to the absence of internal control (human gene), the authors argue that the assay was not designed to confirm *P. falciparum* positivity, rather it was designed to determine hrp2/3 status in P.f positive samples. The authors do not mention which assay was used to detect and quantify Pf of the clinical samples – this needs to be clarified in the methods section, including which qPCR method was used. Adding additional assays to determine the Pf positivity before detecting deletion using current assay will undoubtedly complicate the deletion calling process. Any error introduced during the first assay (to determine Pf positivity) will affect the determination of deletion frequency (under or over estimate) depending how good the assay is.

2. The authors acknowledged this weakness and included a sentence "A sample negative for hrp2/3 and the control gene (e.g. due to pipetting error) will not be classified as carrying a deletion, and will be excluded from any calculations on deletion frequency in a population". If samples negative for both genes are excluded this will under or overestimate frequency estimation. Have the authors encountered such cases? how did they resolve the issue and what was the percentage of the excluded?

3. The dependence for parasite positivity on a different assay prior to using the current ddPCR assay will be a major weakness of the ddPCR assay compared to other qPCR assays that has incorporated parasite detection and quantification within the multiplex assay. The authors need to acknowledge this in the discussion.

4. Regards assay optimisation, the authors used 1.3-49,000 parasites per microliter for assay optimisation/validation but the data has not been shown in the figures cited. The figures for the mixed dilution between 20-1000 (for the mixtures) show that at 20 parasite per reaction (3D7 at 100% or below) there is significant variation and it is not clear how 1.3 parasites per microliter (or as claimed in the abstract 0.33 parasites per microliter) be detected with greater confidence. Since 2 microliter of DNA was added per reaction the minimum parasite per microliter used is 10. The authors need to clarify this.

5. The authors added the requested STDs and CVs for each assay and the CVs look very high for an assay where accuracy was supposed to be a unique selling point. Do the assays need further optimisation to decrease the CVs or the authors need to acknowledge that the limit of detection with greater accuracy (example 3% CV) is higher than they have reported?

6. Detection of hrp2/3 deletion in polyclonal infection is the unique selling point of the ddPCR assay and the authors need to compare the accuracy with other available tools such as qPCR (reference 29 and 31) and genome sequencing (based on coverage difference). If this is not possible, the authors need to acknowledge this weakness in the discussion. The authors rightly pointed out qPCR is generally less accurate (in detecting minor clones) than ddPCR, and the authors have a good opportunity to show that their ddPCR is better than the published qPCR assays in detecting minor clones using data in this study.

*Reviewer #1 (Recommendations for the authors):*

The authors responded adequately to most of the comments. However, there are a few points that the authors need to address:

Major comment (1): In response to the absence of internal control (human gene), the authors argue that the assay was not designed to confirm *P. falciparum* positivity, rather it was designed to determine hrp2/3 status in P.f positive samples. The authors do not mention which assay was used to detect and quantify Pf of the clinical samples – this needs to be clarified in the methods section, including which qPCR method was used. Adding additional assay to determine the Pf positivity before detecting deletion using current assay will undoubtedly complicate the deletion calling process. Any error introduced during the first assay ( to determine Pf positivity) will affect the determination of deletion frequency (under or over estimate) depending how good the assay is.

The authors acknowledged this weakness and included a sentence "A sample negative for hrp2/3 and the control gene (e.g. due to pipetting error) will not be classified as carrying a deletion, and will be excluded from any calculations on deletion frequency in a population". If samples negative for both genes are excluded this will under or overestimate frequency estimation. Have the authors encountered such cases? how did they resolve the issue and what was the percentage of the excluded?

The dependence for parasite positivity on a different assay prior to using the current ddPCR assay will be a major weakness of the ddPCR assay compared to other qPCR assays that has incorporated parasite detection and quantification within the multiplex assay.The authers need to acknowledge this in the discussion.

Major comment (2): While the assays for hrp3 and hrp2 exon 1 are in supplementary figures assay optimisation (limit of detection). For example, the authors used 1.3-49,000 parasites per microliter for assay optimisation/validation but the data has not been shown in the figures cited. The figures for the mixed dilution between 20-1000 (for the mixtures) show that at 20 parasite per reaction (3D7 at 100% or below) show a significant variation and it is not clear how 1.3 parasites per microliter (or as claimed in the abstract 0.33 parasites per microliter) be detected with greater confidence. Since 2 microliter of DNA was added per reaction the minimum parasite per microliter used is 10. The authors need to clarify this.

Major comment (3 and 4): The authors added the requested STDs and CVs for each assay and the CVs look very high for an assay that accuracy was supposed to be a unique selling point. Do the assays need further optimisation to decrease the CVs or the authors need to acknowledge that the limit of detection with greater accuracy (example 3% CV) is higher than they have reported?

Major comment (5): Detection of hrp2/3 deletion in polyclonal infection the unique selling point of the ddPCR assay and the authors need to compare the accuracy with other available tools such as qPCR (reference 29 and 31) and genome sequencing (based on coverage difference). If this is not possible, the authors need to acknowledge this weakness in the discussion. The authors rightly pointed out qPCR is generally less accurate (in detecting minor clones) than ddPCR, and the authors have a good opportunity to show that their ddPCR is better than the published qPCR assays in detecting minor clones using data in this study.

Major comment (6): This is fine.

*Reviewer #2 (Recommendations for the authors):*

The authors have addressed all the previous comments and concerns raised. I am happy with the response from authors and revised manuscript.

---

## [Author Response]

Essential revisionsOverall, there are concerns regards assay robustness and reliability, further details below:1. The assay was designed without an internal (human DNA) control. The absence of internal control makes it difficult to decide whether a negative result is true negative or invalid due to error in DNA extraction, pipetting or a poor sample.

This comment results from a misunderstanding. The assay was not designed to determine whether a sample is *P. falciparum* positive per se (even though *P. falciparum* quantification is built-in the assay). Rather, our assay is designed to determine the deletion status of hrp2 and in samples confirmed to be positive.

We have clarified this as follows (lines 127-129):

“A sample negative for hrp2/3 and the control gene (e.g. due to pipetting error) will not be classified as carrying a deletion, and will be excluded from any calculations on deletion frequency in a population.”

2. In the methods section, the authors mentioned the development of four novel ddPCR assays but they have only shown the validation of hrp2 exon2 and tRNA assays in the result section. Figure 1 only shows the performance of hrp2 exon2 and tRNA assays but not hrp2 exon1 and hrp3.

Assays for hrp2 exon 1 and hrp3 are shown in Figure 1—figure supplement 1 and Figure 1—figure supplement 2.

3. The authors used a "wide range" of parasite densities to optimise and assess the reproducibility of the assays. It is not clear what this range is and the data for reproducibility (variation within replicates) such as STD and CV are missing.

We have added this detail as follows (lines 146 – 147):

“Samples ranging from 1.3 to 49,000 parasites/µL were successfully typed, with samples at densities of >10,000 parasites/µL diluted in H_2_O.”

We added CV on lines 162 – 165:

“The coefficient of variation (CV) across the triplicates was calculated. CV was 17.7% for the hrp2 assay, and 17.1% for the tRNA assay. As no deletions were observed in this sample set, highly similar quantification of hrp2 and tRNA was expected, and CV between hrp2 and tRNA in the same reaction could be calculated. Mean CV between hrp2 and tRNA was 11.8%.

For the *hrp3*/*tRNA* assay, CV was calculated among 96 triplicates. CV was 19.5% for the *hrp3* assay, and 20.7% for the *tRNA* assay. Mean CV between *hrp3* and *tRNA* across all three replicates was 16.8%.”

4. It is not clear why data for repeatability (assay variation between replicates) is not shown.

As stated above, this data has been added to lines 162 – 165.

5. The accuracy of detection of hrp2 clones in polyclonal infection using ddPCR was not validated by any other method – e.g genome sequencing coverage, qPCR etc. This is important to determine if the test if truly fit for purpose.

For clarification, these experiments were conducted on artificial mixtures of laboratory strains, not on field samples of unknown composition. Wild-type (3D7) and Dd2 (carrying hrp2 deletion) were quantified independently by ddPCR, and then mixed in different proportions and at different overall densities.

The reviewer points to an important caveat with these types of data. ddPCR being the most accurate method for DNA quantification, no other method exists to verify the initial quantification of our 3D7 and Dd2 isolates. qPCR is less accurate than ddPCR. Genome sequencing does not yield any data on parasite density.

6. The authors used absolute quantification to measure DNA for both hrp and tRNA, and then used ratio between the two DNA concentrations to determine hrp2 deletion clone in polyclonal infections. This approach assumes that the amplification efficiency for both assays (hrp and tRNA) is the same or similar, but no data has been presented to show this is the case. This is essential to demonstrate

The concept of amplification efficacy does not apply to digital PCR, it only applies to qPCR. In a digital PCR experiment, the number of positive partitions compared to the number of negative partitions determine target density. In the case of low amplification efficacy, the separation between positive and negative partitions might be lower (i.e. they might be closer together). This is one of the main benefits of digital PCR over qPCR for quantification.

Other comments that must be addressed:1. It is unclear why the authors did not compare the ddPCR against the recently described multiplex qPCR assay approaches. There are three qPCR methods that have been recently published and already deployed in field studies and it is essential that the performance of ddPCR is compared against the qPCR methods, including scalability, ease of use, cost and accessibility, particularly in in resource limited laboratories in endemic countries. The authors should include this in the discussion.

ddPCR offers unique benefits, explained above. We believe it would be beyond the scope of this study to compare it against all available other assays.

We have added the following detail to the discussion (lines 368 – 372):

“While digital PCR is less common than nPCR or qPCR, it has been successfully established in malaria-endemic countries. The assay is run in 96-well format; thus throughput is comparable to qPCR, and assay set up is similarly straightforward. While costs of digital PCR instruments and reagents are moderately higher than for qPCR, they have declined recently.”

We are aware of digital PCR being used in Thailand, The Gambia, Ethiopia, and other endemic countries.

2. Line 80-82: reference is missing.

We have added the WHO report for reference.

3. Line 82-84: is it practical to introduce a change in diagnosis policy at sub-national level? Pfhrp2/3-deleted parasites can spread within a country due to test-and-treat policy using HRP2-based RDTs and therefore it is a matter of time before they spread.

This depends on the size of the country, organization of the control program, geographical barriers to transmission, etc. We do not make recommendations on these topics in our manuscript. We have clarified this as follows (lines 84-86):

“The feasibility of such approaches depends on the organization of the national control program, barriers to transmission within-country, and other factors.”

4. Line 95: pfhrp2/3 deletions in multiclonal infections were previously reported (Sepulveda et al. 2018) and reference should be included and the authors need to discuss why their methods is better.

We have added the reference.

Digital PCR offers specific benefits over qPCR, which we have extended in the revised manuscript, but a systematic comparison of all available assays is beyond the scope of this manuscript.

5. Line 96-97: Two more qPCR-based assays for the detection of pfhrp2/3 have been published (Lingard et al. 2020 and Kreidenweiss et al., 2019) with comparable sensitivity and good reproducibility, and the authors need to discuss why their assay is better.

We have added the references.

As stated above, a comparison of all available assays is beyond the scope of this manuscript, and we have not stated that our assay is better than the two mentioned.

6. Line 97-98: It is not clear why the authors suggested quantification is an issue for pfhrp2/3 deletions genotyping. Most HRP2-based RDTs detect as low as 200 parasites per microliter and this parasite density can be accurately detected and quantified using qPCR. The authors need to clarify the issue of pfhrp2/3 deletions genotyping with respect to quantification accuracy.

We have clarified our message as follows (lines 101-109):

“While quantification is accurate in the case of clinical samples that are high density, accurate estimates of the population frequency of deletions might require typing of asymptomatic, low density samples.”

7. 107-110: This claim was not backed up by data.

We have updated the statement as follows:

“The novel assay greatly reduces the number of reactions to be run compared to gel-based assays, showed high sensitivity and accuracy, and can detect the deletion in mixed infections.”

8. Line 123: Provide information about which part of exon the hrp3 primers amplify.

We have added this detail in the methods section (lines 403 – 404):

“The assay amplifies a 101 bp segment in the center of exon 2.”

9. Line 124: The authors say "Upon optimisation of the assay conditions" – where is the data for assay optimisation?

We have clarified that we refer to testing different annealing temperatures only (lines 145 – 146). This is a common procedure when establishing a new assay:

“Upon optimization of the annealing temperature […]”

10. Line 125: The authors mention a wide range of parasite densities but no specific numbers were given.

Please see our response to comment #3 of major comments.

11. Line 128: Figure 1 – how many parasites per microliter is medium and low density? It is not clear whether the hrp2 assay is based on exon 1 or exon 2. Figure for hrp3 in difference parasite density is also missing.

We have added parasite density to all figure legends. We have clarified that Figure 1 shows results of hrp2 exon 2.

12. Line 160: How do the authors came up with the 0.33 parasites per microliter?

We have clarified this sentence as follows:

“Thus, the theoretical limit of detection was 0.33 parasites/µL (3 templates in 9 µL of DNA, of which 6 µL are partitioned into droplets).”

13. Line 138: Are the 248 samples from Kenya cultured? Otherwise, they should be referred to just as samples rather than isolates.

We thank the reviewer for spotting this inaccuracy in our text. We have changes ‘isolate’ to ‘sample’ whenever we refer to field samples, as none of them were cultured.

14. Line 138: The authors used asymptomatic carriers in Kenya to evaluate the reproducibility and limit of detection. Reproducibility and limit of detection are usually conducted using a dilution of a DNA sample with known concentration. The Kenyan samples should be used for validation of the ddPCR assays. In fact, the authors correctly referred the data in figure 2 as "validation". How were the genome copies measured for the Kenyan samples originally and what method was used to quantify them?

This has been added. Please see our response to comment #3 above.

15. Line 139-140: most pfhrp2/3 genotyping assays (nPCR and qPCR) are evaluated based on parasite density (parasites per microliter) and it is difficult to compare ddPCR performance if the parameter is genomes per microliter. The authors need to either use parasite density throughout the manuscript or include a genome to parasite density conversion formula in the analysis section of the methods.

Genomes per µL extracted DNA is another measure for parasite density. To clarify, we have changed the text according to the reviewer’s recommendation and now always refer to ‘parasites/µL’. It is, however, important to remember that there are caveats with this term. Many authors refer to ‘parasite density’, meaning the number of genomes they encounter in extracted DNA, without any control for the amount of DNA lost during extraction. Depending on whether extraction is done from whole blood or filter papers, the volume of blood used for extraction, etc., DNA concentration from the same sample can differ >10-fold (this data stems from a manuscript on a systematic comparison we currently have under review).

We have clarified this in the methods (lines 477 – 481):

“Data is reported as parasites/µL, representing the number of *P. falciparum* genomes per µL eluted DNA. Samples collection and DNA extraction procedures are different among studies (e.g. whole blood vs. filter paper collected, different kits used for extraction). As a result, the conversion from parasites per µL blood to genomes per µL eluted DNA differs among sample sets. For the present study, these differences are not relevant, as for the validation of PCR assays, the number of genomes per µL input DNA is crucial.”

16. Line 139: again the validation was for hrp2 exon-2 and the validation for hrp2 exon-1 and hrp3 are missing.

Please see our response above, in particular Figure 1—figure supplement 1 and Figure 1—figure supplement 2.

17. Line 164: For validating deletions in mixed clones the authors used parasite culture lines 3D7 and Dd2. Why have not the authors used the same parasite lines to assess the reproducibility, repeatability and limit of detection of all individual assays (hrp2 exon-1, exon2, hrp3 and tRNA)?

While we could have prepared several hundred different dilutions (to mimic our Kenya field samples), we do not see a benefit of using cultured parasites rather than field samples to determine reproducibility.

18. Line 164: Clone mixture was done for hrp2 but not for tRNA and hrp3. Since identification of hrp2/3-deleted clone in a multiclonal infection is relative to tRNA detection of mixture should also be validated using tRNA assay.

This experiment has now been done and added as Figure 2—figure supplement 1.

19. Line 270: not clear how 0.33 parasites per microliter is determined.

We have clarified this sentence as follows:

“Thus, the theoretical limit of detection was 0.33 parasites/µL (3 templates in 9 µL of DNA, of which 6 µL are partitioned into droplets).”

20. Line 293: This depends whether the quantification is relative or absolute. In a relative quantification, both targets with similar amplification efficiency are expected to have similar variation of technical replicates and therefore not affecting the mean cycle/quantification threshold value. The authors need to be address this.

As explained above, the concepts of amplification efficacy and relative vs. absolute quantification do not apply to digital PCR. In a dPCR experiment, quantification is always absolute, as it is based on the number of positive partitions. Amplification efficacy is irrelevant as long as a clear separation is seen between positive and negative droplets.

21. Line 296: also due to low parasite density.

Unfortunately, we are not sure what the reviewer means.

22. Line 306: There will be operational challenges to suggest selection of diagnostic tools at sub-national level. Once deletion arise locally it is a matter of time before they spread nationally due movement of people and test-and-treat policy using RDT.

This has been added, please see our answer to this point above.

23. Line 318: systematics surveillance of hrp2/3 deletions should be recommended (if it has not been done yet) before diagnostic policy change.

We have clarified this as follows (lines 357 – 358):

“To determine whether the frequency of the deletion exceeds the threshold of 5%, the WHO recommends systematic surveillance and typing of a minimum of 370 per site [44].”

24. Line 331: Authors mention hrp2 and hrp3 in the conclusion but presented no data about the performance of ddPCR for hrp3.

This has been added.

25. Line 356: The authors stated "The novel primers for exon-2 are located in a region that is deleted in all known deletion variants, thus irrespective of the specific breakpoint, hrp2deletion will be detected" but then they included an assay that targets hrp2 exon 1. Isn't targeting exon-2 only enough if exon-2 is always deleted? This need to be clarified in the materials section as well as discussion.

We thank the reviewer for this comment. We have clarified this in the discussion:

“Though not reported in the literature, deletion of hrp2 exon 1 only, but not exon 2, could result in lack of expression of HRP2. In this scenario, the hrp2 exon 2 assay would not show any deletion, yet HRP2-based RDTs would show a false-negative result. In this study, no discrepancy was observed between deletion status for hrp2 exons 1 and 2. Thus, for future surveillance, it is recommended to only type samples for hrp2 exon 2 and hrp3. In case of negative RDTs despite no hrp2 exon 2 deletion, typing for exon 1 can be done.”

26. Line 359-360: what part of gene does the hrp3 primers target?

Please see our answer above to this question.

27. Line 366-367: Hrp2 probe mutations – Ghana and Mali samples (in the genome database) carry mutation at 5’ end of the probe (C to T). Hrp2 reverse primer targets a region with the same sequence as hrp3 with one nucleotide sequence difference and it is surprising that the primer doesn’t amplify hrp3. In qPCR assay published by Grignard et al. 2020, hrp2 reverse primer with three nucleotide differences amplifies hrp3 and the authors introduced a deliberate mutation at the 3’ end to increase specificity. Vera-Arias and colleagues need to explain how they did not get non-specific amplification (signal) in the hrp3 channel.

Along with other mismatches, when aligned to hrp3, our hrp2 forward primer has a mismatch at the 3’ position. No elongation is possible when a 3’ end mismatch is present. Thus, the hrp2 assays cannot amplify hrp3.

There is no similarity between the hrp3 probe and hrp2, thus no amplification is possible.

Our data (Figure 3) shows clearly that the number of hrp2 and tRNA copies detected is very similar in all samples except where a deletion is present. If unspecific amplification occurred (i.e. amplification of hrp3), the ratio would be systematically higher than 1. This is not observed.

Primers binding to off-target sites (e.g. an *hrp2* primer binding to *hrp3*) can be a problem in qPCR. This is another example while using ddPCR can be beneficial. Only in a few instances will a *hrp2* and a *hrp3* target be partitioned into the same droplet. As long as they are in separate droplets, no off-target binding will happen in droplets containing *hrp2* template.

28. Line 377: why was hrp3 clone detection in mixture strains not done?

This has been added.

29. Line 380: We know that clone mixtures occur at larger ratio difference such as 1:100 and 1:1000 – have the authors considered this? The parasite density used for the mixture is very low (if 500 genome is equal to 500 parasite density per microliter).

We agree with the reviewer and have added the following section (lines 215 – 219):

“The ability to detect a minority clone carrying *hrp2* at a lower frequency (e.g. at 1%) in a sample dominated by a clone carrying the deletion will depend on the overall parasite density. At a density of 1000 parasites/µL 10 droplets are expected to be positive for *hrp2*, thus the wild-type parasite will be detected. At an overall density of 50 parasites/µL, <1 droplet is expected to be for *hrp2*, thus the clone will be missed.”

We have clarified on lines 152-153 that high-density samples (>10,000 parasites) can be diluted before typing.

30. Line 381: how was the ratio calculated? Is it based on standard or relative quantification?

As explained, digital PCR always yields absolute quantification, never relative quantification.

32. Line 384: why was the ddPCR not compared to published qPCR assays?

We believe it would be beyond the scope of this work to compare the assay to all published qPCR assays.

33. Line 429: analysis method for quantification of parasite density (genome) per microliter is missing.

There is no reference to parasite density on line 429, or in this paragraph. In order to avoid any confusion, we have rephrased this sentence as follows (lines 487 – 488):

“If >5 droplets are positive for *tRNA*, the probability of a false-negative result for *hrp2* or *hrp3* (i.e. no positive droplet in a wild-type infection) is less than 1:500.”

34. Line 36 – a "threat" for malaria control.

Thank you, this typo has been corrected.

35. Line 39 – please clarify if LOD parasite/uL is referring to per ul of extract, PCR reaction or original input sample (e.g. whole blood)?

This has been addressed. Please see our response to comment #15 above.

36. Line 60 – how effective are RDTs for diagnosing asymptomatic infections for malaria? They are probably not that sensitive, particularly in the case of COVID-19 detection for asymptomatic cases. Is this the same for malaria?

RDTs are routinely used (and the only tool available)

37. Line 80-82 "At this level, the number of false-negative…." – suggest re-write of this sentence as it is confusing to readers what the authors are intended to compare.

We have rephrased this sentence as follows (lines 81-83):

“At this level, the number of tests that are false-negative tests because of *hrp2* deletion will exceed the number of tests that are false-negative tests because alternative diagnostics offer lower sensitivity.”

38. Line 87-88 "False-negative results could occur when PCR conditions are suboptimal, or when parasite density is low and amplification is stochastic." – a good nPCR assay can potentially amplify up to Ct35. If one can't detect the deletion via nPCR, is the deletion number significant enough? It is unlikely to be picked up by RDTs either at this load. Perhaps this can be confirmed by real-time or digital PCR but how significant is this low parasite load? If the deletion of these genes do not have impact on parasite fitness, then will this low level be of any concern?

Please see the answer to comment #7 above. We have clarified that typing of asymptomatic samples might be required in certain situations.

We agree that nPCR and qPCR can be very sensitive. It is, however, challenging to control for the sensitivity of the *hrp2* and control assays. In the case of nPCR, assays are run in separate tubes, and while a low density sample might result in a band for hrp2, it might not amplify the control gene (as observed in our data from Kenya). In a multiplexed qPCR, slightly different amplification efficiencies of the two assays might result in a positive curve for *hrp2* only, but not the control gene. As explained above, amplification efficiency is not relevant for ddPCR. A wild type sample will always result in positive droplets for *hrp2* and the control gene. The only exception is if parasite density is so low that only DNA templates of one target is added to the PCR. This can occur irrespective of type of PCR, but only in a ddPCR it can be easily controlled for by setting a threshold for the number of droplets positive for the control gene (as we have done in our study).

39. Line 92-04 – will multiple clone (mixed) infection also showed up in RDTs? Might be a good point to add RDTs false negative here to highlight the importance of a good molecular method to detect mixed infections.

We have added this point as follows (lines 97-99):

“While multiple clone infections will result in a positive RDT (if the density of the wild type strain is sufficiently high), their presence can mask the presence of deletions, resulting in an underestimation of the frequency of deletion [30].”

40. Line 166 – "part of all parasites"…the authors meant "a proportion of parasites"?

Thank you, we have corrected this to “when only a proportion of all parasites…”

41. Line 171-174 – in cases where the deletion is hard to observe or densities are too low (small amount of deletion comparing to wild type), do the authors suggest considering running ddPCR in single assay format rather than multiplex to eliminate any resource competitions that might occur in a multiplex reaction?

Resource competition is not a concern in digital PCR, as each partition (droplet) contains only target, either tRNA or hrp2 (with the exception of a small number of droplets that by chance get both targets, pictured in orange in our figures).

42. Line 268 – "An assay" with high sensitivity is required for accurate typing…

Unfortunately, we are not sure to what sentence the reviewer refers. This wording is not present on line 268.

43. Line 275-276 – no deletions were observed by ddPCR but 8% were negative hrp2 but positive for msp2 in nPCR testing. What was done to verify that it was not due to false negative by ddPCR?

This is a misunderstanding. No samples were negative by ddPCR, thus there cannot be any false negative samples.

All samples were run in triplicate for both assays (nPCR and ddPCR), thus avoiding the risk of false-positive results.

44. In Discussion section, it might be useful to mention the cost per sample for ddPCR comparing to nPCR as authors mentioned the reduction in the number of reactions to be run earlier in the article. Cost-effectiveness can usually be a key determinant for many labs to consider when running studies like this.

Please see our response above. We have added a comment to the discussion.

45. Table 1 and supplementary file S1: add quencher (e.g. BHQ1) for Taqman probe sequences.

This has been added.

46. Line 336 -435 – Methods section is difficult to follow. Suggest re-arrange of the sections into Isolates (highlighting how these isolates were determined as positive infections [e.g. by qPCR, microscopy or RDT?]), parasite culture mixtures, ddPCR, nPCR and then data analysis. Since there are many field isolates from different origins, perhaps a table summarising these isolates and positivity rates, how they were detected and special notes on the isolates and how the authors use each isolate to characterise their droplet digital PCR assay. This will make it easier for readers to follow.

We thank the reviewer for this suggestion and have added a table (Table 2).

47. In Supplementary file, please clearly specify the droplet digital PCR instruments, mastermix used, primers and probe (final or working concentrations).

We have added the details of the instruments used and clarified that all primers and probes were added at a concentration of 10 µM.

48. What are the pre-PCR processing for all various field isolates? Were the DNA extracted using different kits and how were they extracted? Please add in manuscript or supplementary methods.

Please see our response to point #15 above. This detail has been added to the methods section.

49. The author also mentioned that this is a validated ddPCR assay. Please consider adding a PCR amplification (droplet scatter) plot showing positive/negative partition with threshold and information on some QC data (for example, positive/negative controls used, acceptable droplet numbers, any rain drop issues from mixed infections etc).

These plots are shown in Figures 1, Figure 1—figure supplement 1, and Figure 1—

[Editors’ note: further revisions were suggested prior to acceptance, as described below.]

Essential revisions:1. In response to the absence of internal control (human gene), the authors argue that the assay was not designed to confirm *P. falciparum* positivity, rather it was designed to determine hrp2/3 status in P.f positive samples. The authors do not mention which assay was used to detect and quantify Pf of the clinical samples – this needs to be clarified in the methods section, including which qPCR method was used. Adding additional assays to determine the Pf positivity before detecting deletion using current assay will undoubtedly complicate the deletion calling process. Any error introduced during the first assay ( to determine Pf positivity) will affect the determination of deletion frequency (under or over estimate) depending how good the assay is.

The reviewer is incorrect that another assay is required prior to the hrp2/tRNA assay to determine *P. falciparum* positivity. Our statement refers to the fact that we are not aware of any study that applied hrp2 deletion typing without any prior step to determine *P. falciparum* positivity, either by microscopy or another PCR assay. If preferred by a researcher, our assay can easily be applied to screen samples for *P. falciparum* and hrp2 deletion at the same time. Of note, none of the frequently used molecular assays (PCR, LAMP, etc.) to determine *P. falciparum* positivity includes a human control. Thus, as with any other assay, if DNA is lost or degraded during extraction, the sample will come up negative (see below for an explanation that this will NOT impact estimates of deletion frequency).

We have clarified this on lines 257-259:

“The assay can be used to screen samples for *P. falciparum* positivity based on the result for *tRNA*, and *hrp2* or *hrp3* deletion in a single assay. Alternatively, it can be used to type samples prior determined *P. falciparum* positive by microscopy or another molecular assay.”

We have added the qPCR for the samples we have screened ourselves to the methods (lines 392-393):

“Samples were determined *P. falciparum* positive either by microscopy (Brazil, Ecuador), 18S qPCR [50] (Ethiopia), or varATS qPCR [51] (Zanzibar, Kenya, Ghana).”

2. The authors acknowledged this weakness and included a sentence "A sample negative for hrp2/3 and the control gene (e.g. due to pipetting error) will not be classified as carrying a deletion, and will be excluded from any calculations on deletion frequency in a population". If samples negative for both genes are excluded this will under or overestimate frequency estimation. Have the authors encountered such cases? how did they resolve the issue and what was the percentage of the excluded?

We disagree, this is a strength, not a weakness. As a reminder, a frequency is calculated as ‘nominator/denominator’. The nominator are all samples lacking hrp2, while the denominator are all samples successfully typed (i.e. positive droplets are seen for tRNA). If a sample is negative, it will be removed from the analysis, i.e. it will not appear either in the nominator nor denominator. Thus, this case will decrease the number of samples typed, but not the proportion.

The reviewer can easily find such cases in our data from Kenya. As described in lines 142-146, we tested the assay on 248 low-density samples in triplicate. Typing was successful in 235/248 samples; thus all frequency calculations were based on a total of n=235 (we did not find deletion in this sample set, but would still make the same calculations if we did).

Now, compare this to a gel-based (nPCR) assay or a qPCR assay where hrp2 and a control gene are typed in separate wells. In case of a pipetting error (i.e. no DNA added to the hrp2 assay), only the control gene might come up positive. Such a sample would wrongly be classified as hrp2 deletion. Our side-by-side comparison of the nPCR to ddPCR has revealed many such cases using the nPCR, resulting in a wrong estimate of hrp2 deletion frequency. The weakness clearly

3. The dependence for parasite positivity on a different assay prior to using the current ddPCR assay will be a major weakness of the ddPCR assay compared to other qPCR assays that has incorporated parasite detection and quantification within the multiplex assay.The authers need to acknowledge this in the discussion.

The reviewer is wrong and there seems to be a fundamental misunderstanding what digital PCR is. There is NO need for another assay prior to our tRNA/hrp2 assay. Both assays are specific for *P. falciparum*. ddPCR is as specific as any other type of PCR. If a sample is negative, no signal will be observed and the samples will be deemed negative. We do not understand where this comment comes from.

4. Regards assay optimisation, the authors used 1.3-49,000 parasites per microliter for assay optimisation/validation but the data has not been shown in the figures cited. The figures for the mixed dilution between 20-1000 (for the mixtures) show that at 20 parasite per reaction (3D7 at 100% or below) there is significant variation and it is not clear how 1.3 parasites per microliter (or as claimed in the abstract 0.33 parasites per microliter) be detected with greater confidence. Since 2 microliter of DNA was added per reaction the minimum parasite per microliter used is 10. The authors need to clarify this.

The 1.3 parasites/uL (or 0.33 parasites/uL) refer to the limit of detection. The 20 parasites/uL refer to the limit of quantification. We have clarified this in the abstract (line 40);

“The deletion was reliably detected in mixed infections with wild-type and *hrp2*-deleted parasites at a density of >100 parasites/reaction.”

5. The authors added the requested STDs and CVs for each assay and the CVs look very high for an assay where accuracy was supposed to be a unique selling point. Do the assays need further optimisation to decrease the CVs or the authors need to acknowledge that the limit of detection with greater accuracy (example 3% CV) is higher than they have reported?

We disagree that the CVs are high. We like to remind the reviewer that we typed low-density, asymptomatic samples where variation in quantification is higher. In fact, such data is usually highly heteroscedastic, as seen in Figure 3. The variation increases fast with lower density and this effect is not fully corrected for by the formula for the CV. This effect is evident when calculating the CV only for samples at a density of >100 parasites/uL. At this density the CV decreases to 11% for the replicates, and to 3-5% for the intra-well comparison (i.e. hrp2 vs. tRNA).

The lower CV within a well than among replicates nicely shows that at lower densities, the major source of variation comes from adding slightly different numbers of template into each replicate just by chance. Maybe this could be further minimized using a pipetting robot, which we have not tested. We are not aware of any molecular assay were a CV of 3% can be achieved at low density.

We have added this to the manuscript (lines 152-156):

“CV was lower among samples of a higher density of >100 parasites/µL (n=126) at 11.5% for hrp2 assay, 10.1% for the tRNA assay, and 4.7% for the within-well *hrp2*/*tRNA* comparison (n=126). The lower CV within wells than among replicates shows that variation in pipetting, resulting slightly different numbers of template added to each replicate reaction, is the main source of variation, while within-well quantification of *hrp2* and *tRNA* is highly accurate.”

6. Detection of hrp2/3 deletion in polyclonal infection is the unique selling point of the ddPCR assay and the authors need to compare the accuracy with other available tools such as qPCR (reference 29 and 31) and genome sequencing (based on coverage difference). If this is not possible, the authors need to acknowledge this weakness in the discussion. The authors rightly pointed out qPCR is generally less accurate (in detecting minor clones) than ddPCR, and the authors have a good opportunity to show that their ddPCR is better than the published qPCR assays in detecting minor clones using data in this study.

We do not agree that this is the unique selling point. We present a very robust method for deletion typing, that we have meanwhile used for hundreds additional samples for follow up studies (see e.g. https://www.medrxiv.org/content/10.1101/2022.04.26.22274317v1 and https://www.medrxiv.org/content/10.1101/2022.04.13.22273827v1). We are in the process of establishing this method in reference laboratories in endemic countries. We do not claim it is the only method that can be used, but we are convinced we present a very useful additional tool for deletion typing.